# Landscape dynamics and the Phanerozoic diversification of the biosphere

Tristan Salles[1,3 ✉], Laurent Husson[2,3 ✉], Manon Lorcery[1,2] & Beatriz Hadler Boggiani[1]

The long-term diversification of the biosphere responds to changes in the physical environment. Yet, over the continents, the nearly monotonic expansion of life started later in the early part of the Phanerozoic eon[1] than the expansion in the marine realm, where instead the number of genera waxed and waned over time[2]. A comprehensive evaluation of the changes in the geodynamic and climatic forcing fails to provide a unified theory for the long-term pattern of evolution of life on Earth. Here we couple climate and plate tectonics models to numerically reconstruct the evolution of the Earth's landscape over the entire Phanerozoic eon, which we then compare to palaeo-diversity datasets from marine animal and land plant genera. Our results indicate that biodiversity is strongly reliant on landscape dynamics, which at all times determine the carrying capacity of both the continental domain and the oceanic domain. In the oceans, diversity closely adjusted to the riverine sedimentary flux that provides nutrients for primary production. On land, plant expansion was hampered by poor edaphic conditions until widespread endorheic basins resurfaced continents with a sedimentary cover that facilitated the development of soil-dependent rooted flora, and the increasing variety of the landscape additionally promoted their development.

The diversity of marine and terrestrial life was assembled over the Phanerozoic eon through complex interplays between biotic controls and abiotic controls[3,4] that are still unclear, although biodiversity patterns over time are fairly well identified from the fossil record[2,5] and mounting evidence from phylogenetics[6,7]. Although both continents and oceans, in the most recent stages of the Phanerozoic, host more species than ever, the monotonic increase of diversity over time in the terrestrial realm[1] contrasts with the more complex evolution of diversity in the oceans[8]. Besides the 'big five' mass extinctions[9], turning points in their progressions also became iconic: Darwin referred to the advent of flowering plants in continents as an abominable mystery; Vermeij[10] coined the term Cenozoic marine revolution. Another enduring puzzle is the late expansion of land plants compared to marine life that rapidly diversified 100 million years earlier. Although the joint effects of biotic and abiotic factors are probably required to explain the biodiversity patterns in time[3] and space[11], a wealth of possible mechanisms have been examined independently. Within this variety, truly independent potential abiotic forcings might have been overlooked, although they are not many and ultimately refer to the physical environment, which couples climatic or geological forcings, suggesting that biodiversity trends could be more comparable for marine and terrestrial life.

Continental drift sets the distribution of landmasses at the surface of the Earth during the Phanerozoic. The changing palaeogeography in turns influences the atmospheric circulation. Both plate tectonics and climate are critical to the development of marine and terrestrial life, by setting the latitude and hours of daylight, temperatures or hydrological cycles. Although these processes are undoubtedly primordial, they do not account for the dynamic evolution of the surface of the Earth, which

should not be regarded as a series of stationary configurations. Reliefs are changing over time and mass transfers are crucial to the expansion of life: both on the continents and in the oceans, nutrient availability is determined by landscape dynamics. Understanding the impact of nutrient fluxes thus requires a comprehensive quantitative approach that we develop herein, leaving aside the role of truly biotic processes.

Here we propose a new method to quantify the global-scale physiographic changes over the Phanerozoic eon, applying the landscape evolution model goSPL[12,13] to a series of global-scale palaeo-elevation reconstructions, consistently tied to a plate tectonic model[14] and to a series of palaeoclimatic reconstructions[15] (Fig. 1). Our approach allows us to jointly quantify the tectonic uplift at long wavelengths and the high-resolution dissection of the landscape (Methods and Extended Data Figs. 1 and 2). Model outputs, including high-resolution topography, continental erosion and sedimentation rates, drainage networks and sediment and freshwater yields to the oceans (all datasets released online[16]), allow estimation of the impacts of surface processes on the physiography of the Earth throughout the entire Phanerozoic (Fig. 1). Sensitivity tests using alternative climatic and tectonic models (Methods) point to spatial variations and differences in the magnitude of erosion rates although global temporal trends remain mostly insensitive (Extended Data Fig. 6a,c).

## Reconstructing sediment flux dynamics

Surface processes are first calibrated using modern estimates of average global erosion rates[17,18] and suspended sediment flux[19,20] (Methods). Then, propagating this parameterization in past times yields temporal

[1]School of Geosciences, The University of Sydney, Sydney, New South Wales, Australia. [2]CNRS, ISTerre, Université Grenoble-Alpes, Grenoble, France. [3]These authors contributed equally: Tristan Salles, Laurent Husson. ✉e-mail: tristan.salles@sydney.edu.au; laurent.husson@univ-grenoble-alpes.fr

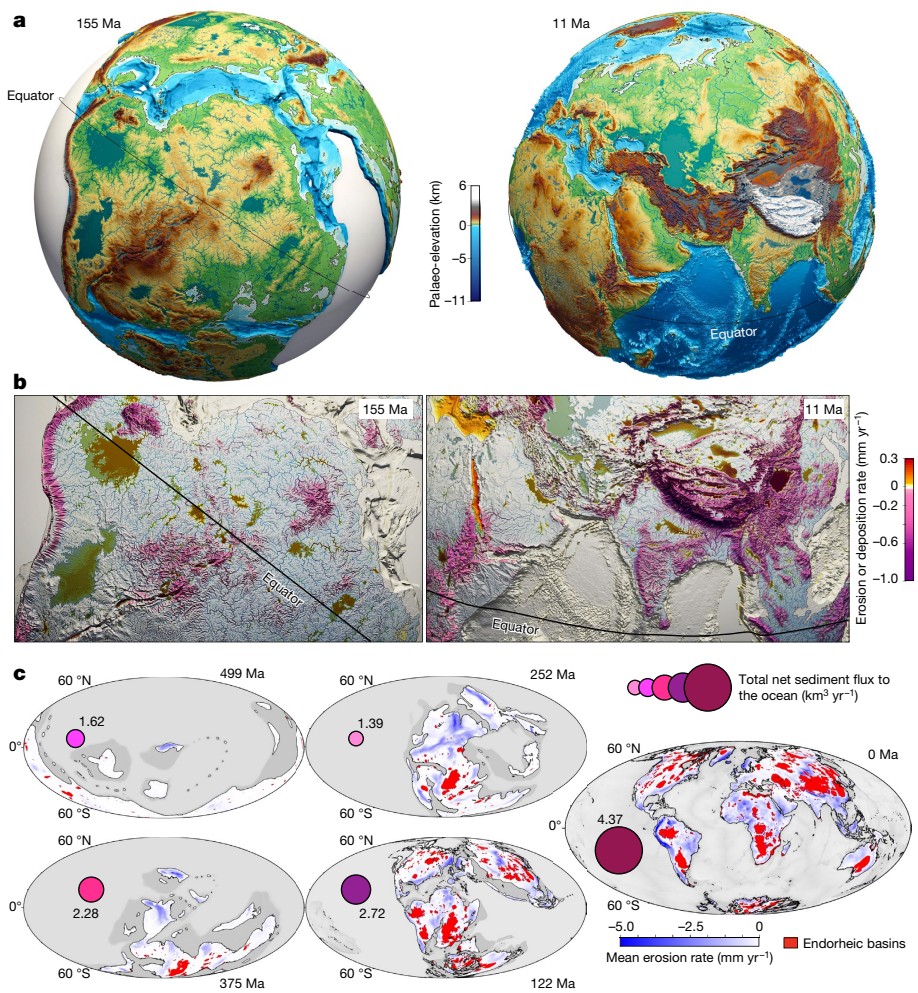

**Fig. 1 | Physiographic evolution and associated patterns of erosion–deposition across the Phanerozoic. a**, goSPL[12,13] simulations showing high-resolution palaeo-landscape, heterogeneous landforms and drainage networks, under the influence of surface processes, at 155 Ma and 11 Ma. **b**, Erosion and sedimentation rates; positive values correspond to deposition in endorheic basins and depressions, and negative ones to erosion across mountain ranges and along major river upstream channels, at 155 Ma and 11 Ma. **c**, Total endorheic sediment coverage since 540 Ma (in red) with cumulative mean erosion rates on continents (blue), and instantaneous global net sediment flux to the ocean for specific time intervals (purple).

trends in bulk sediment transfer (Fig. 2a) that can be tied to continental elevations and surface runoff (that is, precipitation minus evapotranspiration; Extended Data Fig. 6a). Two phases of sustained fast erosion separated by a quieter period mark the long-term Phanerozoic evolution. The Palaeozoic phase relates to an increase in continental runoff from the Silurian period to the Carboniferous period, and to higher reliefs until the assembly of Pangaea during Permian times. Lower continental elevations and more arid conditions prevailed until Pangaea breakup after the Triassic period. During this period, up to 30% of eroded materials were trapped in the terrestrial domain (Fig. 2a). The Meso-Cenozoic phase of erosion, from the Jurassic period onwards, is marked by a more than twofold increase in erosion flux, fostered by higher runoff and by the rising reliefs of the Cenozoic mountain belts. During that phase, most of the sediments are directly transferred to the ocean (continental deposition decreases to about 13% of the erosion flux). Several peaks in erosion flux, coinciding with major orogenic episodes, overprint the low-frequency Phanerozoic trend (Fig. 2a).

By redistributing sediments eroded from the continental reliefs to the oceans, rivers are crucial players in biochemical cycles. However, before extrapolating the model results towards such considerations in deep time, we confront our model predictions with available independent data. First, the geography of model-predicted modern river

outlets and watersheds conforms with actual ones (Extended Data Fig. 3). Likewise, the predicted water discharge and sediment yield for the largest modern rivers compare to current ones[13]. As an example, our predictions of the Amazon River discharge and sediment flux are respectively well within the estimated range (6,591 to 7,570 km³ yr⁻¹)[20] and only about 4% below the sediment production rate inferred from cosmogenic nuclide analysis (about 610 Mt yr⁻¹)[21]. Our model faithfully accounts for the discharge–area scaling relationship between water and sediment flux at the present day[22], and throughout the entire Phanerozoic (Extended Data Fig. 3). Predicted trends of sediment flux compare reasonably well with observations[17], although more closely during the Meso-Cenozoic for which the record is more accurate (Extended Data Fig. 6c). Water and sedimentary fluxes are remarkably anticorrelated over time to the average area of the watersheds (Extended Data Fig. 3c), indicating that large sediment yields are primarily due to small basins characteristic of the heterogeneous landscapes found in tectonically dynamic regions. This is well exemplified by the sharp increase in average drainage basin areas about 240 million years ago (Ma) related to the development of the low-relief landmasses of Pangaea, which closely matches a major decrease in sediment and water flux. Following Pangaea breakup after 200 Ma, water and sediment flux resume owing to wetter climates

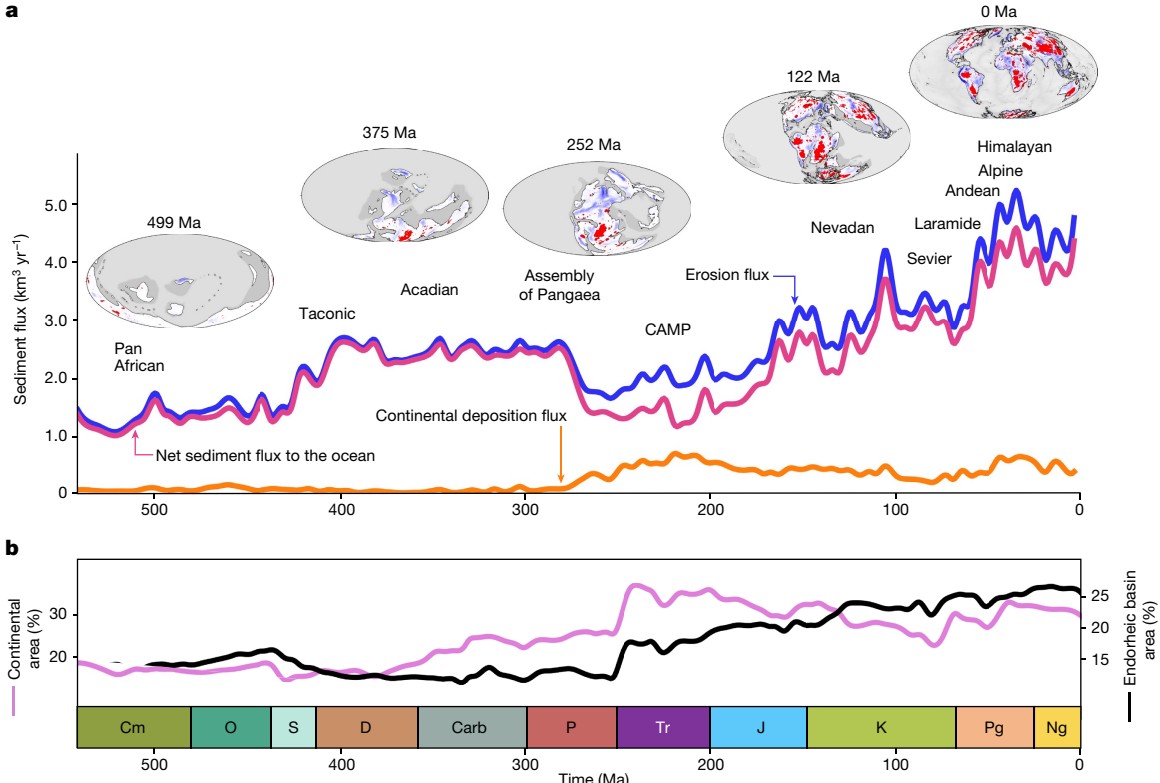

**Fig. 2 | Reconstructed sediment flux and continental sedimentary basin evolution. a**, Simulated global erosion flux, net sediment flux delivered to the ocean and endorheic sedimentation flux; and major orogenic episodes for the Phanerozoic (CAMP, Central Atlantic Magmatic Province). **b**, Global changes in total continental area, and in modelled endorheic basin area. Cm, Cambrian; O, Ordovician; S, Silurian; D, Devonian; Carb, Carboniferous; P, Permian; Tr, Triassic; J, Jurassic; K, Cretaceous; Pg, Palaeogene; Ng, Neogene.

and an overall increase in mean elevation range under renewed plate tectonic activity. We also predict that about 25% of the surface of present-day landmasses is covered by endorheic catchments (Fig. 2b), in agreement with earlier estimates[23]. Additionally, we corroborate the model-predicted sediment flux using the strontium isotopic ratio of seawater for the Phanerozoic, and with a sensitivity analysis based on different palaeo-elevation and palaeoclimate reconstructions (Methods).

## Phanerozoic marine biodiversification

Owing to their quantitative nature, our model predictions provide unprecedented tools to assess the role that physiographic changes might have played in the long-term evolution of the biosphere. During the Phanerozoic, the evolution of the marine biodiversity, derived from palaeontological data[2,5,24], exhibits three major phases (Fig. 3). Following the emergence of the primitive Cambrian fauna, an initial phase of rapid diversification of the Palaeozoic fauna (between the Ordovician and Silurian periods) plateaued up to the Permian period. After a period of lower diversity over the Triassic, marine faunas monotonically diversified and radiated (Extended Data Fig. 10a).

Among the forcings that control the biodiversity, nutrient availability is regarded as one of the most influential environmental drivers because it directly acts on primary productivity within the trophic zone required to sustain marine life[4,24,25] and diversity[26,27]. As nutrient intake by the oceans is primarily related to river runoff, higher erosion rates during orogenic episodes have been proposed as a crucial extrinsic forcing[24,25,28]. Yet inferences between nutrient flux and erosion are to our knowledge only qualitatively assessed, either from the geochemical trends—often matching marine genera to the equivocal $^{87}$Sr/$^{86}$Sr

isotopic ratio of seawater—or by deriving first-order empirical relationships between mountain elevations and sediment flux from major rivers. Our direct quantification of these fluxes over time permits us to alleviate the biases associated with the interpretation of the $^{87}$Sr/$^{86}$Sr ratio of seawater (see Methods) or caused by the default assumption[23] of a linear transfer function between elevation and sediment transport. For example, downstream sediment storage in endorheic basins or reduced precipitation due to orographic shadowing curbs the sediment yield to the oceans, and conversely, small exorheic basins might enhance transport in mid-elevation regions[29].

The reconstructed net sediment flux to the ocean and the total number of marine families are strongly correlated (Pearson coefficient 0.88) and sediment flux variation markedly matches the three main phases that span the Phanerozoic eon (Fig. 3 and Extended Data Fig. 10a). This suggests that nutrient availability is a prime control on marine diversity, providing an explanation for the observed Palaeozoic plateau—as opposed to a continuous increase—and for the Mesozoic marine revolution[10] that sparked biodiversification until the present day, but also for the low-diversity period of Pangaea, when endorheic basins suddenly sequestered a vast amount of sediments over the continents, depriving oceans of about 30% of the nutrient source (Figs. 1c and 2a). The time lag between the Great Ordovician Biodiversification Event and the predicted increase in Palaeozoic sediment fluxes (Fig. 3) could be explained either by uncertainties in the reconstructions of the climate and tectonics, or by the overwhelming effect of climate cooling[30].

Mass extinctions are inescapable attributes of the marine diversity curve[9], which can also be partially matched to the high-frequency variations in the predicted sediment yield to the oceans. Among the big five extinction events, the most pronounced one, during the end-Permian, is associated with the largest drop in sediment flux (Fig. 3).

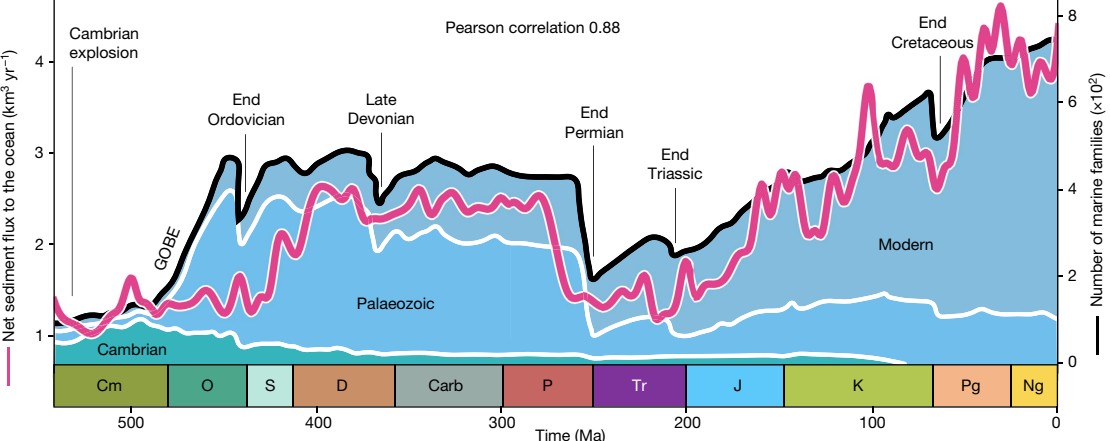

**Fig. 3 | Sediment flux to the oceans and diversity of marine animal families during the Phanerozoic.** Predicted trend in net sediment flux to the ocean (purple curve) and diversity of Cambrian, Palaeozoic and modern marine families (black line indicates total marine families, and Cambrian, Palaeozoic and modern faunas are delimited by white lines, all derived from Sepkoski's compendium[2]; http://strata.geology.wisc.edu/jack/). The Pearson coefficient of 0.88 indicates a strong positive correlation between the two variables. The Cambrian explosion and Great Ordovician Biodiversification Event (GOBE), as well as the big five mass extinction events[9], are indicated.

Besides the end-Ordovician and end-Triassic crises, mass extinction events occurred in the aftermath (about 2 to 15 Myr) of important reductions in the net sediment flux (Fig. 3), caused by either major declines in precipitation rates (Late Devonian) or elevation and palaeogeography (end-Permian), or both (end-Cretaceous; Extended Data Fig. 6a). Conversely, intensified hydrological conditions and weathering, and increases in nutrient discharge, are often considered as major drivers of oceanic anoxia[31] and possibly extinction[32]. Although some congruences between specific Mesozoic anoxic events[33] and peaks in predicted net sediment flux to the ocean can be found in our model, here instead, we posit that sediment shortage—and not excess—more efficiently acts as an essential undermining mechanism before the impact of the compounding processes that ultimately triggered the episodes of mass extinctions (for example, sea-level fluctuations, rapid climatic changes, volcanism and bolide impacts) during Phanerozoic history, as referred to in the press-pulse framework[34].

The identified relationship between marine biodiversity and predicted ocean sedimentary flux could be a direct consequence of the incompleteness and spatial heterogeneity of the fossil record. Many have already raised the issue of preservation bias in the marine palaeobiological record[8,35,36]. If so, the calculated strong correlation would represent an original tool to deconstruct biodiversity curves[37], and computed sediment flux could be used to find under-explored regions with high preservation potentials from the spatiotemporal distribution of simulated palaeo-rivers (Extended Data Fig. 9a). While acknowledging the possibilities for biases in the fossil record[2,5,24], we suggest that the carrying capacity for biodiversity of the oceans is extensively contingent on sedimentary flux and, therefore, on the physiographic evolution of continents. This supports earlier claims that abiotic factors (either environmental[38,39] or related to continental fragmentation and reassembly[40,41]) control speciation and extinction rates. The recently proposed diversity hotspots hypothesis[11] posits that stability in environmental conditions and high continental fragmentation drove the global marine diversity to levels rarely approaching ecological saturation. Our results accordingly support the idea that tectonically driven shifts in palaeogeographies (that is, creation and destruction of geological barriers) and global ocean–atmospheric circulation ultimately affect sediment transport, which in turn modulates the carrying capacity for marine diversity. Our method offers an independent alternative to existing approaches evaluating long-term trends in nutrient flux[24,25,28]; a natural avenue will be to account for the variable lithologies of eroded continental rocks over time and space (for example, large igneous provinces, and continental and volcanic arcs) to precisely quantify the chemical nature of the transferred nutrients (such as silica or phosphorus) that may foster or hinder the development of certain species and trigger evolutionary innovations.

## Phanerozoic terrestrial diversification

Along the same lines, we reappraise the diversification of terrestrial life during the Phanerozoic eon, except that we focus on land plants whose role as primary producers limits the impact of uncontrolled feedback interactions within the trophic chain. For that purpose, we test the possible impact of physiographic changes on vascular plants, by taking as predictors the sedimentary flux onto continents, the gradual spreading of the sediment cover over landmasses and the physiographic diversity of the landscape (Methods).

At first order, the diversification of land plants[1] shows a roughly monotonic increase in the number of species from the Carboniferous onwards (Fig. 4a). Our model results indicate that the sediments accumulated in endorheic basins but that the flux was uneven through time. Owing to the model integration period, the sediment cover is null when the simulation starts, but this does not suggest that no sediment accumulated before the Cambrian period. We however reason that former soft sediments would have turned into barren hard rocks by 450 Ma owing to the limited sediment storage on continents during the Palaeozoic era (Fig. 2a). Sediment flux rapidly rose during the Mesozoic and Cenozoic eras. The good correlation between the sediment flux on continents and the bulk diversity of terrestrial plants (Pearson coefficient 0.67) already suggests that diversification is limited by sediment availability at any time. Moreover, endorheic sediments were mostly preserved after their deposition, thereby increasing the total continental surface covered by sediments (Figs. 1c and 4a). The correlation markedly improves (Pearson coefficient 0.91 (Fig. 4a) and up to 0.96 when limited to the gymnosperms and angiosperms (Extended Data Fig. 10b)) when considering instead that it is the spatial coverage of sediments cumulated over time that limits diversification, by replacing the bare rocks of the substratum with a soil that provides nutrients and moisture to the more specialized plant species that develop over time. Sediment cover is a necessary but non-unique condition for the development of terrestrial plants[42], and for soil and sediment cover to have an effect of diversification, life is required to co-evolve[6,42]. However, after the inception of life, our results suggest that it is the sediment cover that sets the carrying capacity.

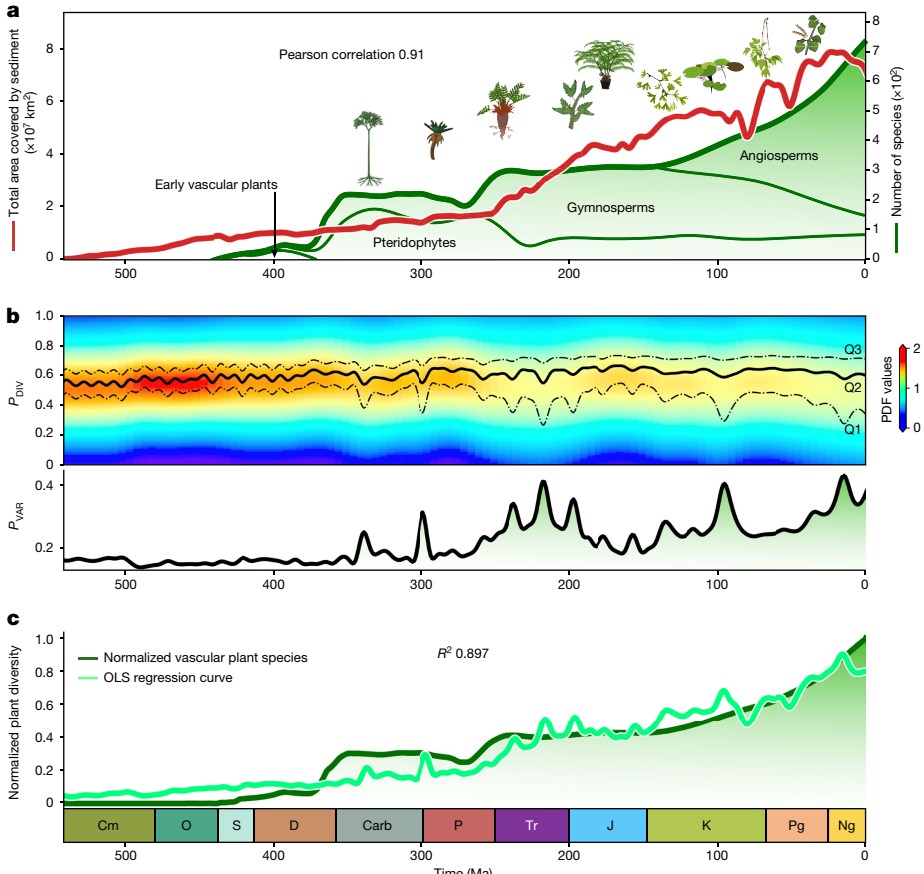

**Fig. 4 | Continental sediment deposition and physiographic complexity, and diversity of vascular plants, during the Phanerozoic. a**, Predicted cumulative area covered by sediments (red curve) and diversity of tracheophyte species throughout the Phanerozoic[1] (green curves). The Pearson coefficient of 0.91 indicates a strong positive correlation between the two main curves. **b**, Top: temporal distribution of the physiographic diversity index ($P_{DIV}$; Methods) with lower (Q1; 25%), median (Q2) and upper (Q3; 75%) quartiles. Probability density function (PDF) is used to estimate the likelihood of having a specific $P_{DIV}$ for each time interval. Bottom: temporal evolution of the physiographic variety

index ($P_{VAR}$ given by the interquartile range of $P_{DIV}$ (Q3–Q1). **c**, Multivariate regression analysis (ordinary least squares (OLS) regression curve) carried out on normalized cumulative area covered by sediments ($S_{COV}$) and physiographic variety ($P_{VAR}$) gives a strong statistically significant relationship ($P$ value < $2.2 \times 10^{-16}$). The resulting regression equation is defined by $0.019 + 0.27P_{VAR} + 0.61S_{COV}$. Analysis of variance shows a high dependence of plant diversity dynamics on these abiotic parameters ($R^2 \approx 0.9$). Botanical icons by Rebecca Horwitt, available at full size and open access from https://sites.psu.edu/rhorwitt/.

A further incentive to diversification comes from the increasing physiographic variety of the landscape (Fig. 4b) since the Carboniferous. Whereas the mean physiographic diversity ($P_{DIV}$; Methods) varies only moderately throughout the Phanerozoic eon, the physiographic variety ($P_{VAR}$; Methods) varies strongly. From the Triassic to the Cretaceous period, and during the Cretaceous and Cenozoic, the variety of the landscape increases at times when the overall rate of diversification accordingly increased (Fig. 4). By offering new habitats, periods of increased topographic heterogeneity have been identified as drivers of diversification at the regional scale[43,44]; our results qualify this observation and indicate that the impact is in fact global, but also that it is the variety of the landscape—from low and high diversity—that promotes the overall diversification of terrestrial plants (Fig. 4c).

More insights can be gained by scrutinizing the evolution at genus level (Fig. 4a). During the early Palaeozoic era, continents covered less than 20% of the Earth surface with restricted endorheic basins (about 17% of emerged lands; Fig. 2b) and sparse continental deposition, hampering both soil production by physical and chemical weathering and preservation. Irrespective of climatic or biological factors, these poor edaphic conditions are suitable only for non-vascular plants that inhabit a variety of substrates (including bare rock) and access nutrients directly from meteoric waters and leachate[4,7]. Early vascular

plants radiated during the Devonian period, with the development of arborescent species and seed plants[4,6]. Our reconstructions show that at that time, the increased global sediment flux (Fig. 2a) was not stored in endorheic basins (Fig. 4a), and that the physiographic variety was low (Fig. 4b). The low diversification of early vascular plants on land was thus driven only by species adaptation and climatic forcing rather than by geomorphological changes[45,46]. This is corroborated by the increasing tolerance of plants to water stress and seasonality[47] associated with the colonization of diverse environments[6,48] at that time, feeding back on the landforms they live on[6].

The diversity of land plants steeply increased only during the Late Devonian epoch with the rapid rise of pteridophytes and gymnosperms, but diversity quickly plateaued until the mid-Permian. As the total sediment coverage of landmasses stalled during that period (Fig. 4a) despite sustained erosion flux (Fig. 2a), we suggest that the diversity of terrestrial plants was further hampered by the limited expanse of favourable edaphic conditions.

Over the Permian and Triassic, pteridophytes were superseded by gymnosperms that further radiated (Fig. 4a). At that time, the Pangaea supercontinent gathered more emerged lands than at any other time in the Phanerozoic (Fig. 2b). Sediment-covered surface areas also rapidly increased owing to the development of large endorheic basins (up to 20% of the continental surface; Fig. 2b), fed by a sustained flux of

sediments from the high relief of the widespread circum-Pangaean orogenies[49] (Fig. 4a). The massive development of these reliefs is also associated with an increase of physiographic variety (Fig. 4b). The emergence of these conditions, which would favour the diversification of deep-rooting plants across a diverse range of physiographic environments, coincided with the development of gymnosperms (Fig. 4) that dominated terrestrial floras by the end of the Triassic[4].

Gymnosperm diversity continued to rise during the early phases of Pangaea breakup before levelling off during the Jurassic and Cretaceous, along with a decrease in both continental deposition flux (Fig. 2a) and physiographic variety (Fig. 4b), as well as a relatively steady sediment coverage (Fig. 4a), which all restrained the favourable rejuvenation of continental surfaces. Gymnosperms were superseded by angiosperms that diversified at unprecedented rates at least from the Cretaceous onwards (Fig. 4a and Extended Data Fig. 10b), although the timing remains largely controversial[50,51]. Common explanations invoke their efficient cross-pollination strategies and high growth rates[52,53]. Yet the period was also marked by extensive orogenic phases in North and South America and Eurasia (Fig. 2a). At that time, erosion resumed in the reliefs along with renewed endorheic deposition in the lowlands, and the overall physiographic variety further increased (Fig. 4b). The new diverse niches that developed in this heterogeneous topography, along with quickly expanding, nutrient-rich continental surfaces, could have promoted the fast radiation of angiosperms.

## Conclusion

Our study shows a remarkable congruence between the Phanerozoic landscape dynamics and the diversification of both marine life and terrestrial life. Earlier work already identified elements of this, but the analyses remained fragmentary[25,53–56], considering isolated pieces of the environmental puzzle: climatic, geological or biotic. Here we suggest that the evolution of continental physiography—as set by the interplay between the geosphere and the atmosphere—determines nutrient availability, and that it is a crucial limiting factor in both the marine realm and terrestrial realm, as important as intrinsic biotic processes[9,10,53], or extrinsic processes such as the climatic control[46] or plate tectonics[41]. In the oceans, riverine sedimentary flux directly sets nutrient availability for primary productivity. In continents, nutrient availability is tuned by endorheism, by rerouting the sedimentary flux and gradually varnishing their surfaces with a sedimentary cover, which facilitates the development of more specialized species. The relative effects of physiographic diversity and erosion rates on diversification are difficult to discriminate, but we suggest that the variety of the physiography further adjusts the effect of endorheism by tessellating the landscape.

The modality of sediment routing implies that diversification is simultaneously detachment limited (the sediment flux should be enough to sustain diversification) and transport limited (sediment storage in continents may, in an extreme case, starve marine life while instead feeding terrestrial life, or vice versa). The Phanerozoic trends of marine and terrestrial diversity highlight these regimes: marine diversity directly scales with sediment flux and is thus dominantly detachment limited. Land plant diversity is instead transport limited: its onset occurred much later than that of marine diversity and exploded only once endorheism efficiently resurfaced continents with sediments. Overall, physiographic changes determine the carrying capacity of both the oceans and the continents.

We anticipate that these findings, together with the released sets of physiographic descriptors at a high spatial resolution for the past 540 Myr (ref. 16), will invite more quantitative reappraisal of the interactions between the solid Earth and the atmosphere, hydrosphere and biosphere. For instance, our current approach conveniently reduced the problem to the temporal dimension by extracting spatially averaged metrics but ignores the spatialization of diversification events.

A thorough palaeogeographical analysis of diversification events[56] in both continents and oceans is now permitted thanks to these reconstructions. Sensitivity tests, which illustrate how denudation rates scale with climate reconstructions and endorheic sediment storage is chiefly controlled by palaeo-elevation reconstructions, will allow further testing of our hypothesis.

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

## Methods

### Global landscape model

Here we use goSPL[12,13], an open-source scalable parallel numerical model that simulates landscape and sedimentary basin evolution at the global scale (resolution about 5 km). It accounts for river incision and soil creep, both considered as the main drivers of long-term physiographic changes. goSPL also tracks eroded sediments from source to sink considering alluvial and marine deposition, sediment compaction and porosity change, and could be used to reconstruct basin stratigraphic records. To evaluate these processes on landscape dynamics, different forcing conditions could be imposed from spatially and temporally varying tectonics (both horizontal and vertical displacements) to multiple climatic histories (for example, precipitation patterns and sea-level fluctuations). The model's main equation, the continuity of mass, has the following common form:

$$\frac{\partial z}{\partial t} = U + \kappa \nabla^2 z + \epsilon P^d (PA)^m \nabla z^n \qquad (1)$$

for which changes in surface elevation $z$ with time $t$ depend on tectonic forcing $U$ (rock uplift rate, in metres per year), hillslope processes for which $\kappa$ is the diffusion coefficient (set to 0.5 m$^2$ yr$^{-1}$)[57] and fluvial processes defined using the stream power law. $m$ and $n$ are dimensionless empirical constants (set to 0.5 and 1), $\varepsilon$ is a precipitation-independent component of erodibility (set to $4.0 \times 10^{-7}$ yr$^{-1}$ on the basis of the choice of $m$), and $PA$ is the water flux combining upstream total area ($A$) and local runoff ($P$) obtained from palaeoclimate mean annual precipitation minus evapotranspiration[57]. In our formulation, the weathering impact of runoff and its role on river incision enhancement is incorporated by scaling the erodibility coefficient with local mean annual runoff rate with a prefactor $d$ (a positive exponent estimated from field-based relationships[58] and set to 0.42). It follows from equation (1) that deposition in flat plains or along gentle slopes is null. However, it simulates continental deposition in depressions and endorheic basins.

In goSPL, erosion occurring in upstream catchments is linked to basin sedimentation through a multiple-flow-direction algorithm that routes both water and sediment flux towards multiple downstream nodes, preventing the locking of erosion pathways along a single direction and helping the distribution of the corresponding flux in downstream regions. To solve the flow discharge globally ($PA$), we use a parallel implicit drainage area (IDA) method[59,60] in a Eulerian reference frame, expressed in the form of a sparse matrix composed of diagonal terms set to unity and off-diagonal terms corresponding to the immediate neighbours of each vertex composing the spherical mesh. The solution of the IDA algorithm is obtained using the Richardson solver with block Jacobian preconditioning[59], both available in PETSc[61]. Continental erosion and sediment transport solutions follow a similar approach[60]. Some of the main advantages of goSPL lie in its design of implicit and parallel solutions of its constitutive equations[60], making it possible to increase the model stability even with large time steps, and to scale the simulation run time over hundreds of CPUs.

**Palaeo-elevation and precipitation forcing.** To reconstruct the past physiography, goSPL relies on time-evolving boundary conditions—climatic and palaeogeographic—that are used to compute the interplay between the solid Earth and the climate. To reconstruct high-resolution palaeo-elevations throughout the Phanerozoic, we use a series of 108 maps from the PALEOMAP palaeogeographic atlas[14] ranging from the Holocene epoch to the Cambrian–Precambrian boundary (541 Ma). These palaeo-maps are defined at approximately 5-Myr intervals, and each of them is represented as a regular grid with a resolution of 0.1° × 0.1° (approximately 10-km cell width at the Equator). These palaeogeographic maps were initially based on information related to lithofacies and palaeoenvironmental datasets[62] and supplemented

and refined for more than 40 years with regional palaeogeographic atlases[14,63]. We acknowledge that these maps bear some uncertainties and controversial aspects. For example, the very early Andean rise to their modern elevations consequently precociously lowers the predicted bulk sediment flux during the Neogene period while diversification continues to increase (Figs. 3 and 4). It is worth noting that even though the PALEOMAP dataset forms the basis of this study, from a methodology standpoint, other datasets[64–66] could be used.

To simulate riverine processes, the palaeo-precipitation dataset used was generated using a variant, HadCM3BL-M2.1aD (ref. 15), of the coupled atmosphere–ocean–vegetation Hadley Centre model. This climate model also uses the PALEOMAP Atlas[67] but at a lower resolution (3.75° × 2.5°). The reconstructed palaeo-precipitation regimes are obtained for each individual palaeo-elevation map after running the climate model for at least 5,000 model years to reach a dynamic equilibrium of the deep ocean[15]. In addition to palaeo-elevation grids, there are two additional time-dependent boundary conditions that were set in the climate model: the solar constant; and the atmospheric $CO_2$ concentrations. Regarding the last condition, two alternative $CO_2$ estimates are proposed[15] and we chose the palaeo-precipitation and evapotranspiration maps generated from the set of HadCM3 climate simulations using the $CO_2$ local weighted regression curve from ref. 68.

From the palaeo-elevation and palaeo-runoff maps, we generate the input files for goSPL by resampling the global temporal grids on an icosahedral mesh composed of more than 10 million nodes and 21 million cells (corresponding to an averaged resolution of about 5 km globally—about 0.05° resolution at the Equator). Inspired by techniques used in palaeoclimate modelling[15,67], we designed an approach to achieve a dynamic equilibrium (erosion rates balance tectonics; equation (1) and Extended Data Fig. 2) under steady boundary conditions (rainfall, tectonic uplift and erodibility). For each individual time slice, we run two sets of simulations over 168 CPUs to estimate their corresponding physiographic characteristics and associated water and sediment dynamics (Extended Data Fig. 1). A first simulation is carried out over an interval of 2 Myr under prescribed elevation and runoff conditions and simulates landscape evolution and associated water discharge and sediment flux assuming no other forcing. Under this setting, the role of surface processes is not counterbalanced by tectonics, and they excessively erode the reconstructed elevation, trimming part of the major long-lived orogenic belts and upland areas, and causing extensive floodplains. The resulting elevations are then corrected by assimilating the palaeo-elevation information[13]. Model predictions account for landscape evolution, and at this stage already contain a more detailed representation of terrestrial landforms (for example, canyons, valleys, incised channels and basins to cite a few) than the initial palaeo-elevation (Extended Data Fig. 2a). To preserve these morphological features during the correction step, high-amplitude and high-frequency structures are removed using a combination of moving average windows (ranging from 0.5° to 2°) that conserves the global distribution of the initial palaeo-elevation with minimal change of its hypsometry (≤0.5%; Extended Data Fig. 2c). We then derive a tectonic map (uplift and subsidence rates) by computing the local differences between the palaeo-elevation values and the adjusted ones.

A second simulation starts with previous palaeo-elevation and runoff conditions and additionally accounts for tectonic forcing. This simulation runs until dynamic equilibrium is reached (that is, erosion rates compensate tectonic uplift rates) within the first million years of landscape evolution (Extended Data Fig. 2c). The outputs of this second simulation are then used to evaluate water and sediment flux for the considered time slice, as well as the major catchment characteristics (river networks, drainage areas, and erosion and deposition rates).

The parametrization of equation (1) is obtained by calibrating its variables with modern estimates of average global erosion rates[18] (mean value of 63 m Myr$^{-1}$ with a standard deviation of 15 m Myr$^{-1}$; Extended Data Fig. 6c) and of suspended sediment flux from the BQART model[19,20]

(corresponding to 12.8 Gt yr$^{-1}$). Following calibration, we predict an average present-day global erosion rate of 71 m Myr$^{-1}$, and a sediment flux of 12.15 Gt yr$^{-1}$ (assuming an average density of 2.7 g cm$^{-3}$). On the basis of the multiple-flow IDA approach used to integrate runoff over upstream catchments[59] (IDA algorithm), we also extract the spatial distribution of the largest water discharges and sediment flux (Extended Data Fig. 3a) and their respective basin characteristics.

## Sr isotopic variations from mantle origin

We use the geochemical archive of oceanic sediments to test the validity of the model predictions. The $^{87}Sr/^{86}Sr$ isotopic ratio of seawater (Extended Data Fig. 4) reflects the balance between continental weathering and mantle dynamics (hydrothermalism at mid-ocean ridges and weathering of island arcs and oceanic islands)[69,70], making it a classic proxy to diagnose the relative importance of geodynamic and climatic forcings through time[71,72].

From the measured isotopic budget of the ocean (O), present-day low $^{87}Sr/^{86}Sr$ ratios (about 0.703) are produced from mantle sources (M), whereas high ratios (about 0.713) come from continental runoff (CR, measured from main rivers worldwide)[69]. As a result, the strontium isotope oceanic mass balance has the following form:

$$\left[\frac{^{87}Sr}{^{86}Sr}\right]_O = \xi\left[\frac{^{87}Sr}{^{86}Sr}\right]_M + (1-\xi)\left[\frac{^{87}Sr}{^{86}Sr}\right]_{CR}$$
$$\left[\frac{^{87}Sr}{^{86}Sr}\right]_M \approx 0.703; \quad \left[\frac{^{87}Sr}{^{86}Sr}\right]_{CR} \approx 0.713 \tag{2}$$

in which $\xi$ represents the mass fraction of the Sr coming from the mantle ($Q_M/(Q_M + Q_S)$ with $Q_S$ being the predicted net sediment flux to the ocean derived from our simulation) and the flux of mantle origin ($Q_M$). At the present day, $Q^0_M$ is given by the percentages defined above:

$$Q^0_M = rQ^0_S \quad r = 0.41/0.59 \tag{3}$$

Assuming that weathering scales with erosion rates, our reconstructed global net sediment flux to the ocean ($Q_S$) offers an independent alternative to existing approaches evaluating Sr flux from tectonic origin and could be used to infer past tectonic activity[73–76]. Under this assumption, differences ($\Delta(^{87}Sr/^{86}Sr)$) between our predicted Sr ratio and the one from the geological record[39] would reflect the dynamics of the Earth's exogenic system, with positive $\Delta(^{87}Sr/^{86}Sr)$ values corresponding to periods of higher tectonic activities relative to the present day and negative ones coinciding with more quiescent periods. We then use the isotope oceanic mass balance to independently derive the mantle contribution to the $^{87}Sr/^{86}Sr$ ratio, relying on our estimates of terrigenous flux and on the observed $^{87}Sr/^{86}Sr$ ratio of seawater. We crudely use the Sr isotope oceanic mass balance to estimate the Sr flux of mantellic origin ($Q_M$; Extended Data Fig. 4) in the mass fraction $\xi$:

$$Q_M = Q_S \frac{\left[\frac{^{87}Sr}{^{86}Sr}\right]_{CR} - \left[\frac{^{87}Sr}{^{86}Sr}\right]_O}{\left[\frac{^{87}Sr}{^{86}Sr}\right]_O - \left[\frac{^{87}Sr}{^{86}Sr}\right]_M} \tag{4}$$

Note that we assume that the isotopic ratios $\left[\frac{^{87}Sr}{^{86}Sr}\right]_{CR}$ and $\left[\frac{^{87}Sr}{^{86}Sr}\right]_M$ remained equivalent to those of the present day. Whereas the Sr ratio from the mantle budget might change marginally ($^{87}Sr/^{86}Sr$ about 0.703 for mid-ocean ridge hydrothermal and about 0.7035 for island arcs and oceanic islands[69]), the contribution of the continental crust to the Sr ratio is highly dependent on the type of weathered materials[70] ($^{87}Sr/^{86}Sr$ about 0.708 for limestones, compared to about 0.721 for silicates[69]).

We find that the contributions of mantle and terrigenous sources relative to those of the present day covary at long wavelengths, with two periods of reinforced influx from both sources separated by a quieter period during Pangaea. This trend mirrors the Wilson cycle. It indicates that the periods of high erosion, coinciding with periods of continental aggregation and contraction, increased elevation and wetter climates, also match periods of reinforced tectonic activity. Seafloor expansion is faster during periods of continental dispersal, and the total length of ridges increases during breakup[77]. As the mantle input to the $^{87}Sr/^{86}Sr$ ratio of seawater is partially driven by seafloor kinematics, we find several similarities between the predicted mantle Sr flux and subduction rates[72,74,76] that can be taken as a proxy for oceanic spreading rates; Extended Data Fig. 4). Notably, over the past 250 Ma, we deduce that mantle fluxes were low during Pangaea, and subsequently increased during Pangaea breakup; flux decreased during the late Palaeogene, mirroring the decrease in crustal production rates in the Atlantic and Pacific oceans[78] and the consumption of the East Pacific ridge.

Our model predictions of sediment flux compare at first order to the observed increase in the $^{87}Sr/^{86}Sr$ ratio of seawater over the past 150 Myr (Extended Data Fig. 4). Assuming that weathering scales with erosion rates, it corroborates the first-order impact of Cenozoic orogenesis[79] on the Sr ratio. Likewise, over the entire Phanerozoic, short-lived (20–40 Myr) increases in predicted erosion flux can explain the increase in the $^{87}Sr/^{86}Sr$ record during major orogenic phases[80], whereas the long-term variations of the record can be at first order explained by the coevolution of the terrigenous and mantle sources during the Wilson cycle. These results indirectly substantiate our model predictions of sediment flux to the oceans.

## Limitations and sensitivity

In goSPL[12], erosion is defined using a first-order parametrization of the physics at play (equation (1)), which captures the long-term, large-scale landscape evolution[22,57,81]. More sophisticated treatments directly linked to sediment transport theory and incorporating different erosional behaviours have been proposed (for example, by accounting for mobile sediment and bed resistance to erosion, or using different formulations of sediment transport based on transport-limited equations)[82–84].

In addition, the erodibility parameter does not consider spatiotemporal variations that could be induced by environmental (for example, temperature gradients and seasonal precipitation), geological (for example, soil composition and fault-induced bedrock weakening) or biological (for example, plant root growth and soil microbial activity) mechanisms[6,85–89]. Instead, we assume uniform erodibility across all continents. Accounting for variable lithologies in model space could be achieved by assigning an erodibility prefactor depending on the surface rock composition in the stream power law term of equation (1) by an erodibility prefactor depending on the type of surficial lithology classes (typically with values ranging between 1.0 and 3.2; ref. 90). However, this approach would require global palaeo-lithological surficial cover data that are difficult to obtain when looking into deep geological times. Although we do not account for seasonality variations, the weathering impact of precipitation and its role on river incision enhancement is incorporated by scaling the erodibility coefficient with the local mean net annual precipitation rate[58]. One could also incorporate the effect of temperature on weathering of rocks according to rock composition by scaling the erodibility coefficient using the palaeoclimate temperature distribution. Such refinement possibilities are many, and although in principle desirable to better reproduce observations, adding those would necessarily add poorly controlled degrees of freedom in the parameterization, and lead to illusory enhanced predictive capabilities.

By design, our simulation is sensitive to both the climatic and palaeo-elevation reconstructions. Although other palaeogeography reconstructions exist[62,64–66], many are restricted to specific geological intervals

and, to our knowledge, are not tied to a series of palaeo-precipitation maps for the entire Phanerozoic. Consequently, we chose to evaluate model sensitivity on palaeo-elevation using a single time slice (Aptian period about 120 Ma) by comparing our predicted sediment flux with a different set of palaeogeography and palaeoclimate reconstructions[91]. The results highlight several differences at the regional scale (Extended Data Fig. 5). For instance, a more humid equatorial climatic zone in the second reconstruction[92] induces higher erosion rates on the northern and central part of Gondwana. The palaeo-elevation differences also redefine the drainage network and the volumes of sediment transported to the oceans or stored in continental basins. This is the case in Antarctica and on the eastern part of Eurasia where we note higher erosion rates or an increase in terrestrial sediment accumulation depending on the considered palaeogeography. Despite these local variations, those disparities become more tenuous when evaluating the global response. As an example, the percentage of endorheic basins varies from 24 to 26.5% between the two simulations and the net sediment flux to the ocean changes from 2.72 to 2.26 $km^3 yr^{-1}$ (Extended Data Fig. 5). This suggests that although regional differences exist and if the imposed forcing conditions are not too dissimilar (both in terms of palaeoclimates and palaeogeographies), the reported global evolution and global trends that are presented in the study should remain relatively unchanged between reconstructions.

To evaluate the model sensitivity to palaeo-runoff, we ran a full series of simulations throughout the Phanerozoic. The palaeoclimate reconstructions from ref. 15 have been carried out with two different $CO_2$ conditions (the atmospheric concentrations from ref. 68 and a smoothed curve), but the mean global continental runoff remains very similar in both cases; Extended Data Fig. 6a) and should only very marginally change our results. Instead, we opted for the recent Phanerozoic palaeo-precipitation reconstruction from ref. 93 that was run at 10-Myr intervals using the PALEOMAP palaeogeographic atlas[14] with a much higher resolution (1°) than those of ref. 15. The release from ref. 93 does not contain the evapotranspiration time slices, and we could therefore use the total palaeo-precipitation only as a proxy for runoff. Global mean continental runoff from ref. 93 exhibits a similar temporal trend to the ones from ref. 15 but, because evapotranspiration could not be accounted for, with higher values (about 0.3 m $yr^{-1}$ on average over the Phanerozoic; Extended Data Fig. 6a). As erosion scales with runoff (equation (1)), this inflated runoff directly affects the global net sediment flux to the ocean (about 0.84 $km^3 yr^{-1}$ on average) and to a lesser extent the continental deposition flux (<0.1 $km^3 yr^{-1}$ on average; Extended Data Fig. 6b). The spatial distributions of these two runoff scenarios and their relative impact on denudation rates show substantial spatial differences over time (Extended Data Fig. 7). At the continental scale, the higher resolution in ref. 93 should better account for the control of topography on the spatial variability in precipitation. For example, we note at 40 Ma (Extended Data Fig. 7) the orographic control of the Himalayas on the regional rainfall regime with its implications on erosion and deposition.

The sensitivity analysis provides two important pieces of information. First, both simulations show similar responses in terms of global sediment flux and denudation rates (Pearson correlations of 0.94 and 0.92, respectively). This suggests that irrespectively of the chosen palaeoclimatic reconstruction, our interpretation of the relationships between predicted sedimentary flux and biodiversity still holds. Second, the runoff has a stronger effect on the amplitudes in net sediment flux to the ocean and denudation rates when considering similar palaeo-elevation reconstruction (Extended Data Fig. 6b). Instead, differences in palaeo-elevations affect continental sediment cover and sediment accumulation in endorheic basins (Extended Data Fig. 5).

Another limitation of our approach is to propose a hypothesis by comparing time series of mean model outputs with independent variables, but similar trends could possibly be expected for any model that accounts for plate tectonics—with essentially two cycles of continental

aggregation and dispersal over the Phanerozoic eon—and subsequent climate reconstructions. The highly relevant correlations that we found can be however advocated to hierarchize these studies. As plate tectonics—and the breakup of Pangaea in particular—form the cornerstone of such studies[11,40,41], we thus compared continental fragmentation[41] with Phanerozoic marine diversity, which yields only moderate correlations (Pearson coefficients of 0.46 with number of marine families[2]; 0.54 to 0.58 with the number of genera[2,5]; Extended Data Fig. 8).

## Physiographic diversity and variety

To evaluate the relationships between physiography and the Phanerozoic climate and tectonics, we define a unique continuous variable (named the physiographic diversity index) that encapsulates several of the reconstructed morphometric attributes. Simulation outputs are first interpolated from the icosahedral mesh on a regular 0.05° grid (open-access online release, HydroShare[16]).

First, we quantify palaeo-landforms by calculating the topographic position index on each cell $i$ ($TPI_i$) that measures the relative relief[94]:

$$TPI_i = z_i - \sum_{k=1}^{n} z_k / n \quad TPI_S = 100 \times (TPI - \overline{TPI}) / \sigma_{TPI} \qquad (5)$$

in which $z_i$ is the considered elevation at cell $i$ and $z_k$ is the mean of its surrounding cells ($z_k$), with $n$ being the number of cells contained inside an annulus neighbourhood. Topographic position is an inherently scale-dependent calculation[95]. To circumvent this problem, TPI is computed considering two scales of observation, a fine one ranging between 0.05° and 0.15° and a coarser one from 0.25° to 0.5°. As elevation is generally spatially autocorrelated, TPI values increases with scale, making it difficult to compare both scales of observation directly. To overcome this issue, we calculate a standardized $TPI_S$ (equation (5)), in which $\overline{TPI}$ is the mean over the entire grid and $\sigma_{TPI}$ is its standard deviation[96] (top map in Extended Data Fig. 9a).

We also extract the slope and water discharge for each time slice (bottom maps in Extended Data Fig. 9a). Note that we selected these three variables—TPI, slope and discharge—as morphometric indicators of the physiographic diversity but other ones such as aspect (that is, the direction of maximum gradient), and erosion and deposition rates could equally be added[97]. From these continuous variables, we then derive categorical variables by defining seven categories from the slope, five from the water flux and ten from the $TPI_S$ (Extended Data Fig. 9a,b and Supplementary Table 1; ref. 98).

From the categorical variables, we quantify the physiographic diversity index $P_{DIV}$ (Supplementary Fig. 1a,b) using Shannon's equitability (continuous variable [0,1]), which is calculated by normalizing the Shannon–Weaver diversity index ($d_{SW}$):

$$d_{SW} = - \sum_{k=1}^{\mathcal{C}} p_k \ln(p_k) \quad P_{DIV} = d_{SW} / \ln(\mathcal{C}) \qquad (6)$$

with $p_k$ being the proportion of observations of type $k$ in each neighbourhood and $\mathcal{C}$ being the number of categorical variables (here $\mathcal{C} = 3$). The physiographic diversity index is calculated at each location for the 108 reconstructed time slices spanning the Phanerozoic (Supplementary Fig. 1c). The variations of $P_{DIV}$ can be described either spatially for a given period (Supplementary Fig. 1c) or across time (Supplementary Fig. 2). As it characterizes how landscape complexity evolves over the geological timescale[99], this index can be used to infer the role physiography, and changes thereof, might have played on species migration, dispersal or isolation at both global and continental scales. Biodiversity richness emerges from many abiotic and biotic interactions; however, the role physiography might have played has often been overlooked[100]. The high-resolution global maps generated from our simulation could be used as an additional palaeoenvironmental layer in mechanistic eco-evolutionary models[101,102].

To compare the temporal evolution of physiographic diversity with biotic[2,5] or abiotic geochemical and geophysical proxies, the mean value of $P_{DIV}$ is insufficient, as it ignores the variety of landforms, which is better accounted for by the probability density function: the probability density function can be usefully reduced to a unique scalar—physiographic variety—given by the interquartile range $P_{VAR} = Q_3 - Q_1$ (that is, the range between the first quartile ($Q_1$; 25%) and the third quartile ($Q_3$; 75%); Supplementary Fig. 2c).

As an example, we observe an increase in physiographic diversity of South America from the Upper Cretaceous to the Palaeocene related to the Andean mountain building period (Supplementary Fig. 2b,c) and concomitant with microflora diversity patterns in northern South America[103] and plant diversification in Patagonia[104]. We also note periods of low $P_{VAR}$ values around 50 Ma and 28 Ma related to the intermittent development of internal seas or lakes in the central part of the Amazon basin. Periods of increasing $P_{VAR}$ follow each of these episodes, suggesting that the Pebas basin could have acted as a permeable biogeographical system favouring biotic exchange and adaptation in the region[105].

## Reporting summary

Further information on research design is available in the Nature Portfolio Reporting Summary linked to this article.

## Data availability

The PALEOMAP Paleodigital Elevation Models for the Phanerozoic from ref. 14 can be downloaded from https://doi.org/10.5281/zenodo.5460860 (last accessed 3 October 2023). Palaeoclimatic maps from the HadCM3BL-M2.1aD model[15] are available from the Bristol Research Initiative for the Dynamic Global Environment website at https://www.paleo.bristol.ac.uk/resources/simulations/. Palaeoclimate simulations from ref. 93 are made available in the original article, as referenced. All palaeo-elevation reconstruction maps[16] for the Phanerozoic generated in this study are available from HydroShare: http://www.hydroshare.org/resource/0106c156507c4861b4cfd404022f9580.

## Code availability

The scientific software used in this study, goSPL[12], is available from https://github.com/Geodels/gospl and the software documentation can be found at https://gospl.readthedocs.io. We also provide a series of Jupyter notebooks used for processing the datasets and model outputs that can be followed to reproduce some of the figures presented in the paper, which can be accessed from https://github.com/Geodels/paleoPhysiography. The open-source python interface for the Generic Mapping Tools (https://www.pygmt.org) was used for global two-dimensional map visualization and the three-dimensional global maps in Fig. 1 were created with the open-source Paraview software (https://www.paraview.org).

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

**Acknowledgements** This work was principally supported by the Australian Research Council DARE Centre grant IC190100031. This research was undertaken with the assistance of resources from the National Computational Infrastructure, which is supported by the Australian Government, and from Artemis HPC Grand Challenge supported by the Sydney Informatics Hub at the University of Sydney. In addition, we acknowledge C. Nielsen, M. Laugié and A. Lettéron for providing the Aptian palaeo-elevation and palaeo-precipitation grids used in Extended Data Fig. 5c. The manuscript was improved by the reviews and suggestions of R. Martin and A. Pohl.

**Author contributions** T.S. and L.H. contributed equally to this work; they conceived the study and conducted the numerical experiments. M.L. and T.S. worked on the physiography index definition and interpretations. L.H. and T.S. derived the strontium budget. All authors analysed the endorheic basin evolution. L.H. and T.S. wrote the initial draft and all authors reviewed the manuscript.

**Competing interests** The authors declare no competing interests.

**Additional information**
**Correspondence and requests for materials** should be addressed to Tristan Salles or Laurent Husson.

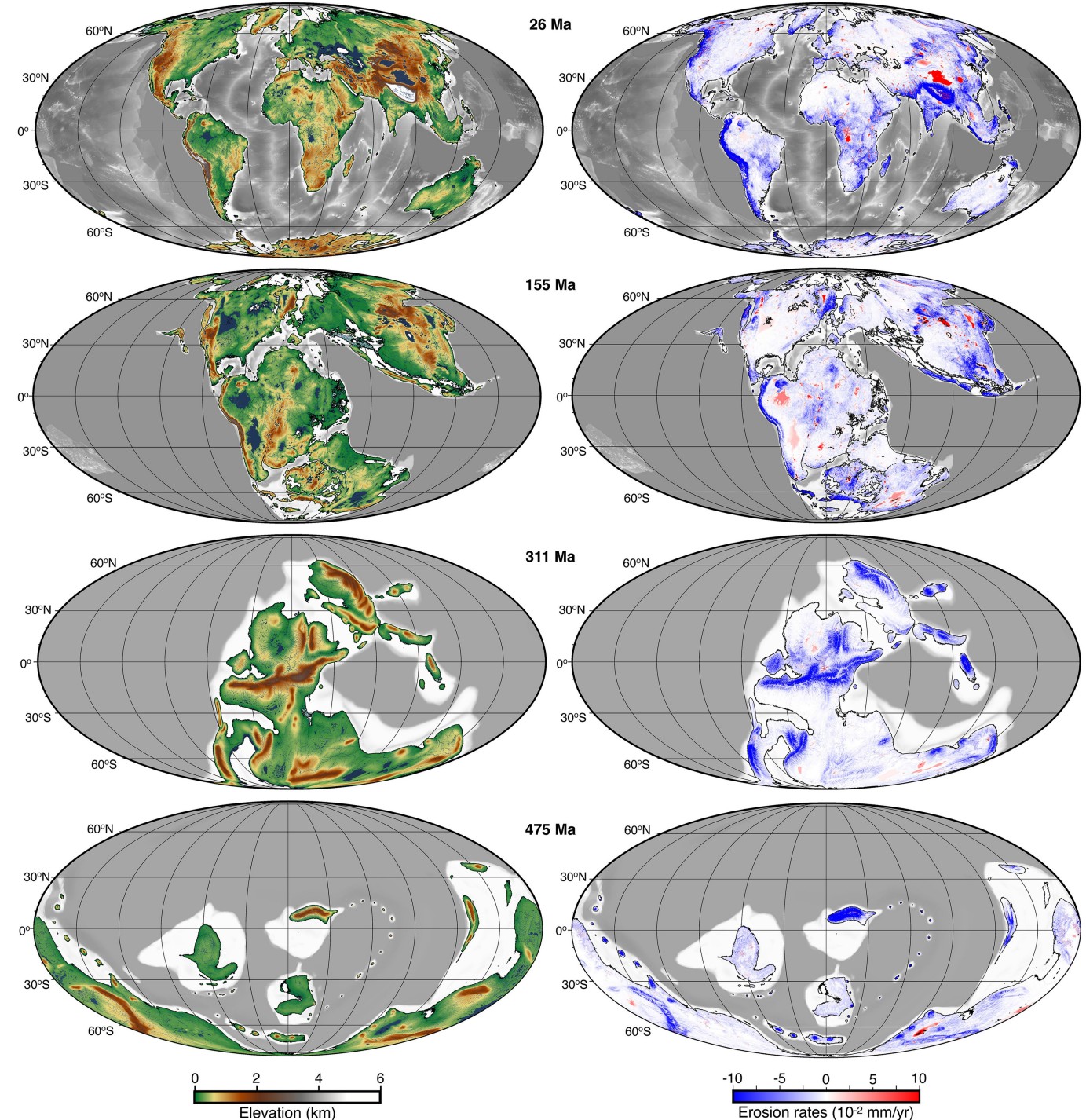

**Extended Data Fig. 1 | Global scale Phanerozoic landscape evolution model.**
Left panels represent simulated physiographies for 4 time-slices accounting for surface processes impact and highlighting continental topography and associated river networks (dark blue). Right panels show associated erosion/deposition rates (blue/red respectively) for the considered time-slices.

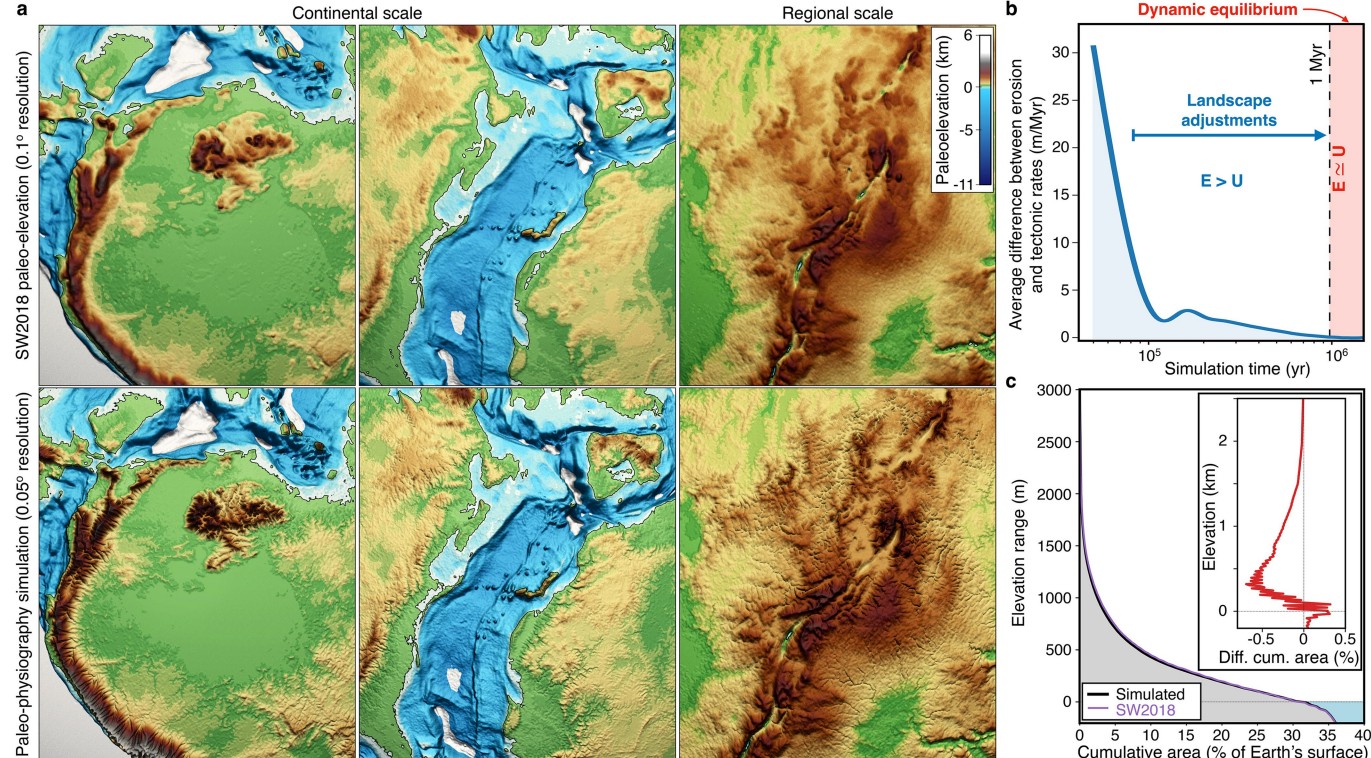

**Extended Data Fig. 2 | Comparisons between predicted elevations and corresponding paleo-elevation map. a**. Top 3 panels show the input elevation conditions for 155 Ma at 0.1° resolution (SW2018[14]) and bottom ones represent model outputs (0.05° resolution), highlighting the geomorphological imprints of surface processes on the landscape. **b**. Temporal change between imposed tectonic rates from corrected topography and erosion rates at 155 Ma (blue curve). This curve is used to estimate when dynamic equilibrium conditions have been reached. **c**. Corresponding continental hypsometric curves for the given paleo-elevation at 155 Ma (purple) and simulated one (black). Red curve in the inset shows the differences between the two hypsometric curves.

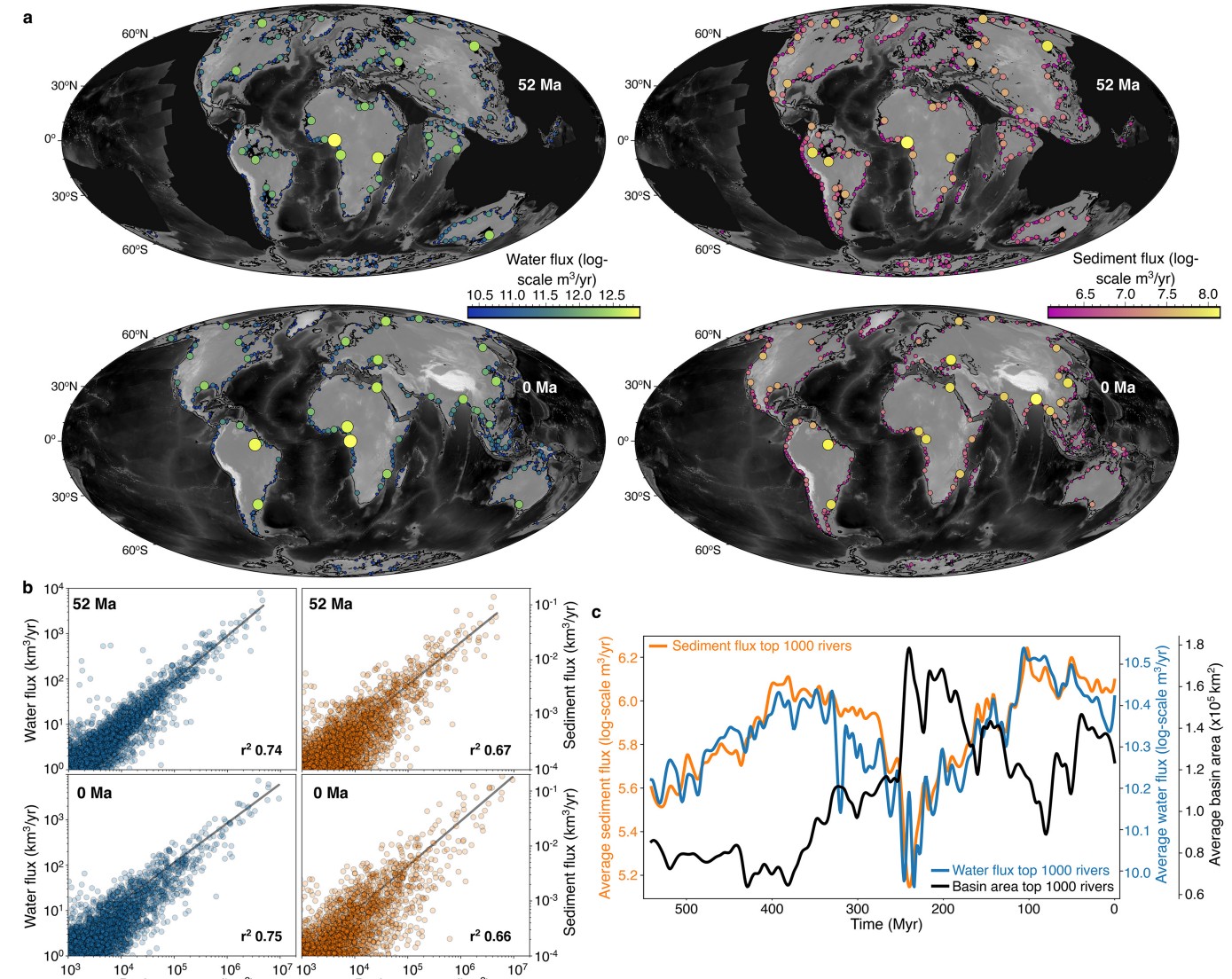

**Extended Data Fig. 3 | Drainage patterns evolution at catchment scale.**
**a**. Distribution of water and sediment flux for the 500th largest rivers at 52 and 0 Ma. **b**. Log-log plots of modelled water discharge and sediment flux against basin drainage area (blue and orange circles respectively). Power law curve fitting and is represented by the black lines. **c**. Phanerozoic evolution of average water discharges (blue), sediment flux (orange) and basin areas (black) for the top 1000th largest rivers.

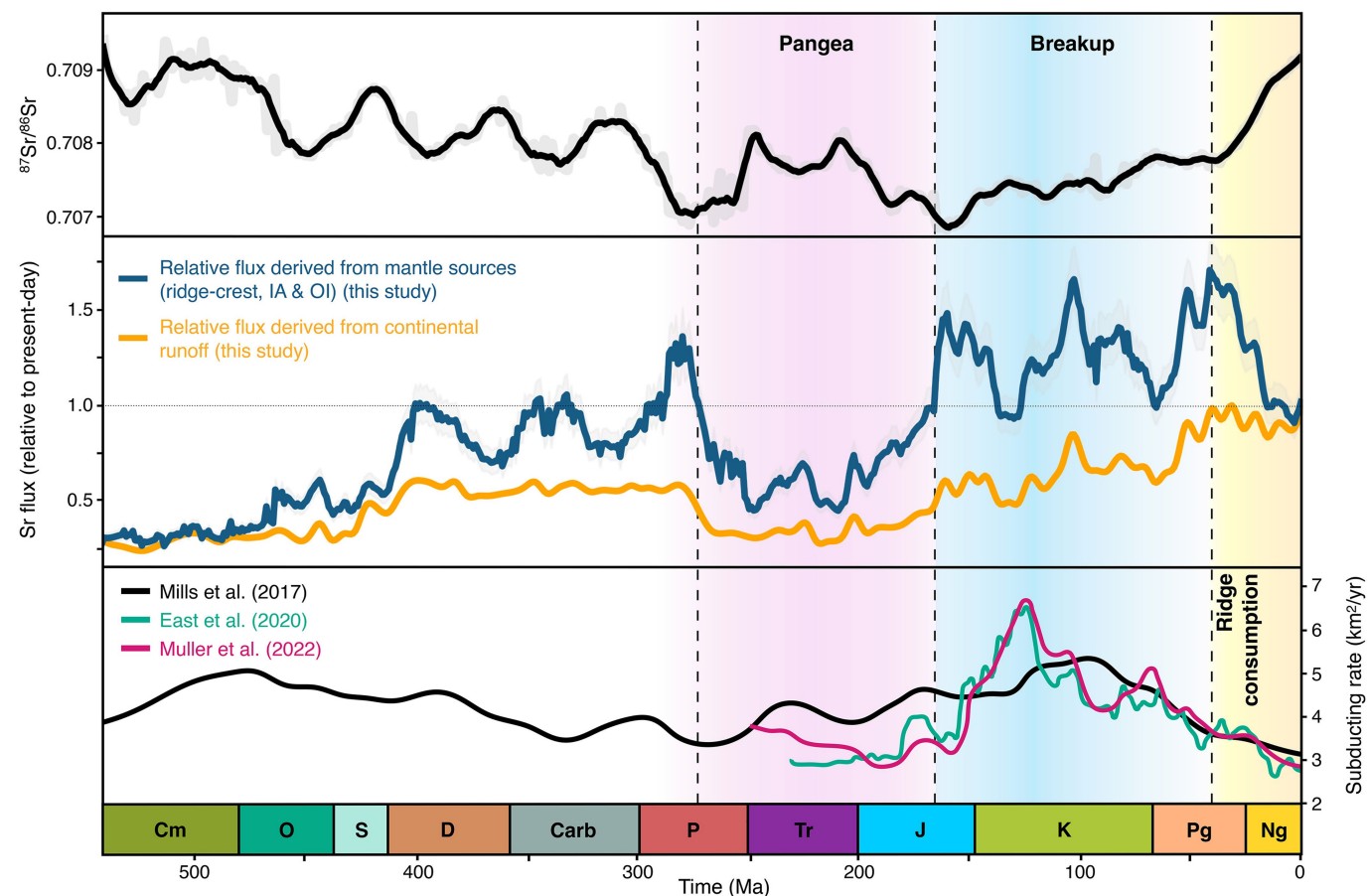

**Extended Data Fig. 4 | Strontium signal from mantellic source and continental runoff.** Top panel shows Strontium isotopic ratio of seawater for the Phanerozoic[39] (grey line represents the data from[39] and the black line shows the corresponding least square regression using SciPy Savitzky-Golay filter). From the oceanic Strontium mass balance (Methods) and using present-day weighted average inputs[69] (i.e., 0.7136 ± 0.0002 from continental crust and 0.7029 ± 0.0003 from mantle sources), the relative strontium mantle budget is derived. The mantellic flux associated to ridge-crest, island arcs (IA) and oceanic islands (OI) underground alteration is predicted in the second panel based on the flux obtained from continental runoff (grey area shows the extent of the flux based on the range ±0.0003 from[69]). Reconstructions of mean crustal destruction rates from recent studies[72,74,76] are presented in the bottom panel.

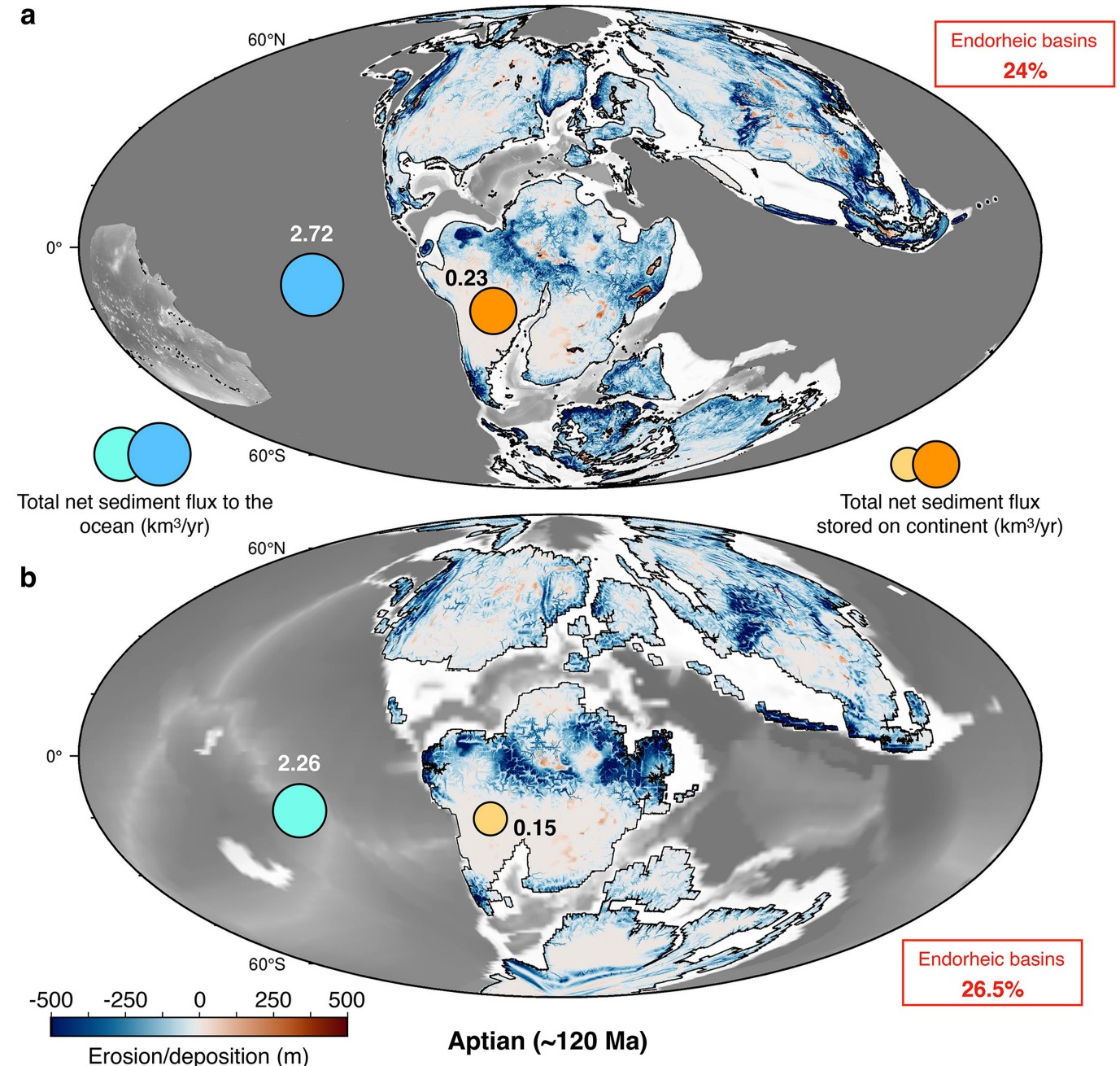

**a**

60°N

Endorheic basins
**24%**

0°

2.72

0.23

60°S

Total net sediment flux to the
ocean (km³/yr)

Total net sediment flux
stored on continent (km³/yr)

**b**

60°N

0°

2.26

0.15

60°S

Endorheic basins
**26.5%**

-500    -250    0    250    500

Erosion/deposition (m)

**Aptian (~120 Ma)**

**Extended Data Fig. 5 | Impact of paleogeographies and paleoclimates on sediment flux dynamic for the Aptian period (~120 Ma).** Predicted continental erosion and deposition maps computed using the paleo-elevation of Scotese & Wright[14] and its associated paleo-precipitation from Valdes et al.[15] in **a** and using the continental reconstruction and paleo-precipitation provided by Nielsen, Laugié and Lettéron and obtained from the IPSL-CM5A2 paleo-climate model from Sepulchre et al.[92] in **b**. Estimated sediment flux delivered to the ocean (blue gradients circles) and stored on continents (orange circles) are shown as well as the percentage of endorheic basin area for each simulation.

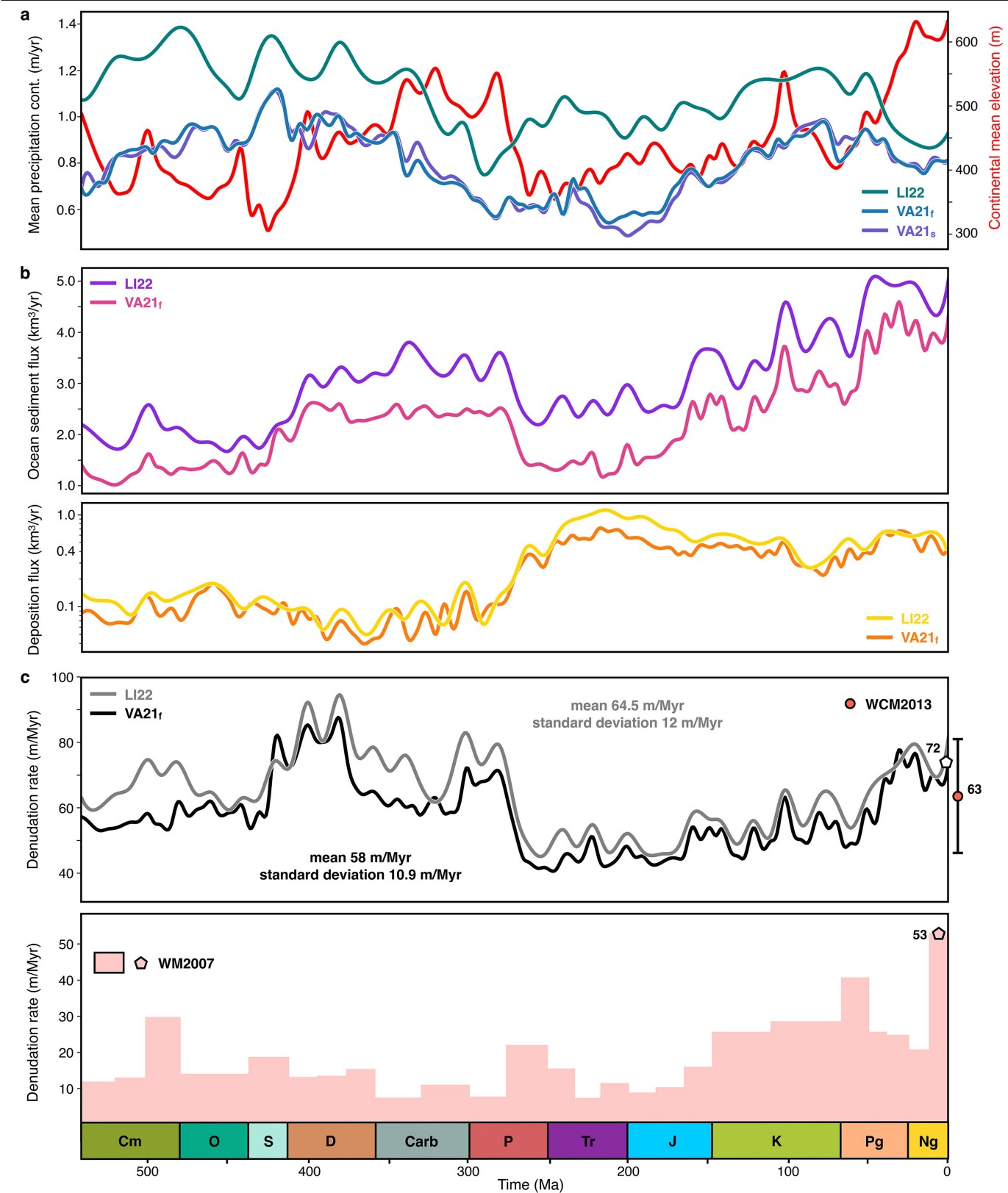

**Extended Data Fig. 6 | Comparisons between paleo-precipitation scenarios and global sediment flux for the Phanerozoic. a**. Evolution of continental paleo-precipitation for three scenarios (Li et al.[93] (LI22) in teal, Valdes et al.[15] using the CO_2 curve from Foster et al.[68] (VA21_f) in blue, and a smoothed CO_2 curve (VA21_s) in purple) and paleo-elevation[14] (red). **b**. Estimated net sediment flux to the ocean (fuchsia and magenta curves) and continental deposition flux (yellow and orange curves) for two climatic scenarios (LI22[93] and VA21_f[15] respectively). **c**. Global average denudation rates estimated for the simulated paleo-precipitations (LI22 and VA21_f – grey and black curves) against estimated rates from preserved sediments (pink, WM2007[17]) and recent rates (red, WCM2013[18]).

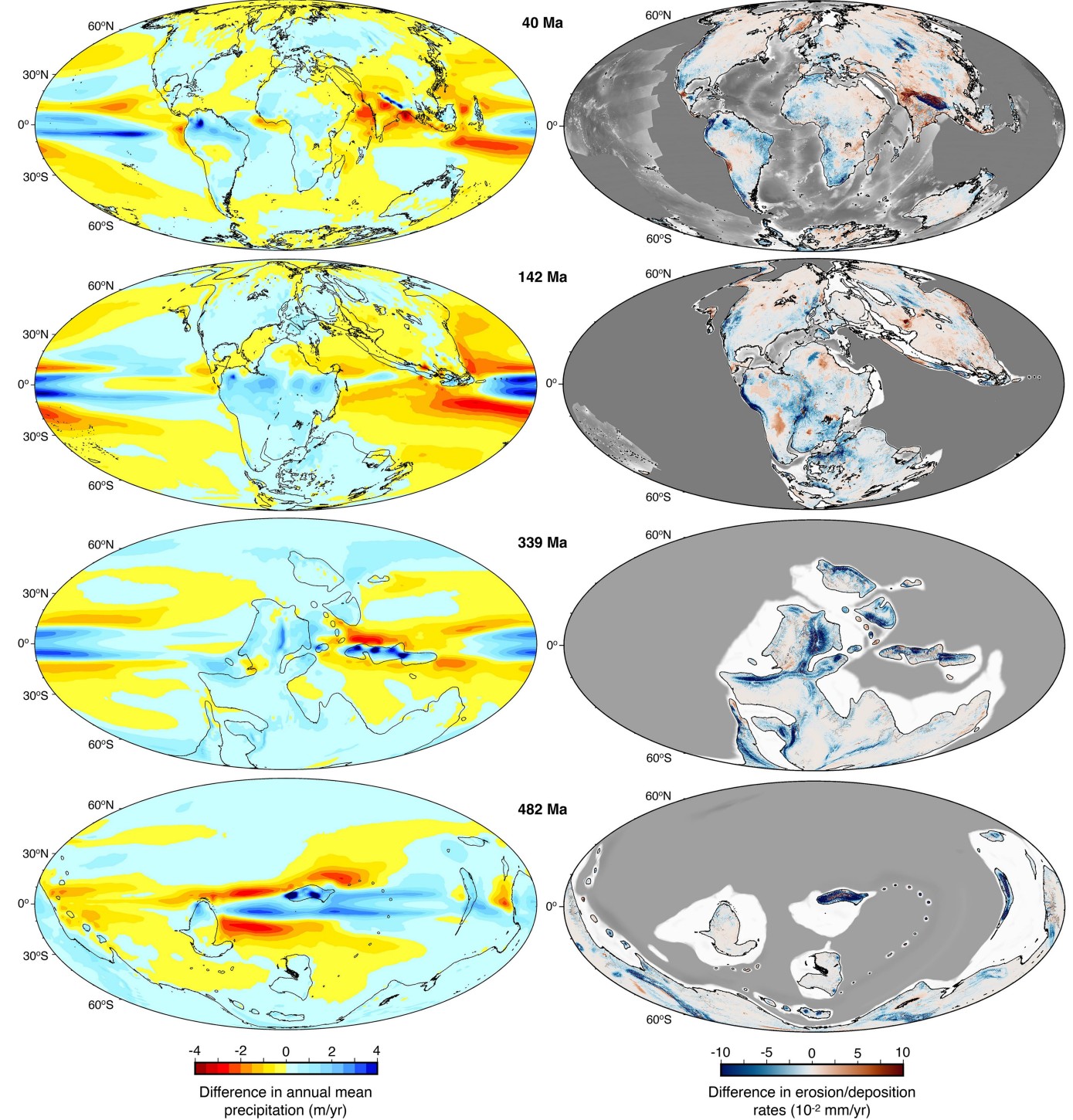

**Extended Data Fig. 7 | Influence of paleo-precipitation scenarios on spatial distribution of denudation rates.** Differences in annual mean precipitation between the Phanerozoic paleo-climate model of Li et al.[93] and the one from Valdes et al.[15] using the $CO_2$ curve from Foster et al.[68] are presented on the left (drier predictions in Li et al.[93] range from yellow to red and wetter ones from cyan to dark blue). Simulated differences in erosion and deposition rates between the two scenarios using goSPL[12] show higher erosion with the Li et al.[93] simulation in blue and higher deposition in red.

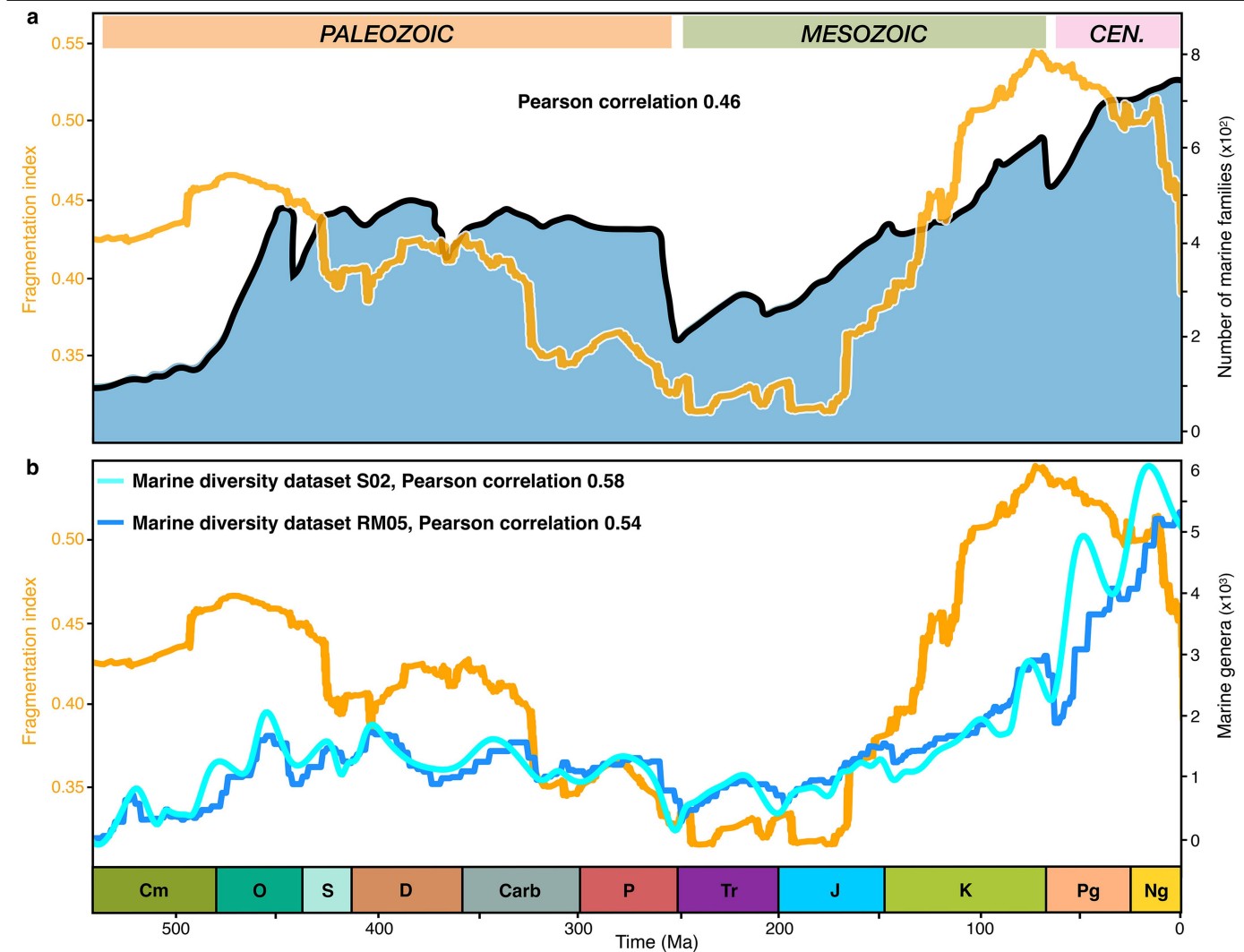

**Extended Data Fig. 8 | Plate tectonics and the Phanerozoic marine biodiversity. a**. The index of continental block fragmentation derived from Zaffos et al.[41] (orange curve) with value close to 1 indicating no plates are touching and a value of zero corresponding to contiguous continental blocks arranged in a single mass. Pearson correlation (0.46) indicates a positive but moderate relationship with the number of marine families (black line from Sepkoski's dataset[2]. **b**. Correlations between the fragmentation index and the total number of marine genera from Sepkoski[2] (SO2) and Rohde & Muller[5] (RM05) show a slightly stronger but still moderate positive trends (Pearson values of 0.58 and 0.54 respectively).

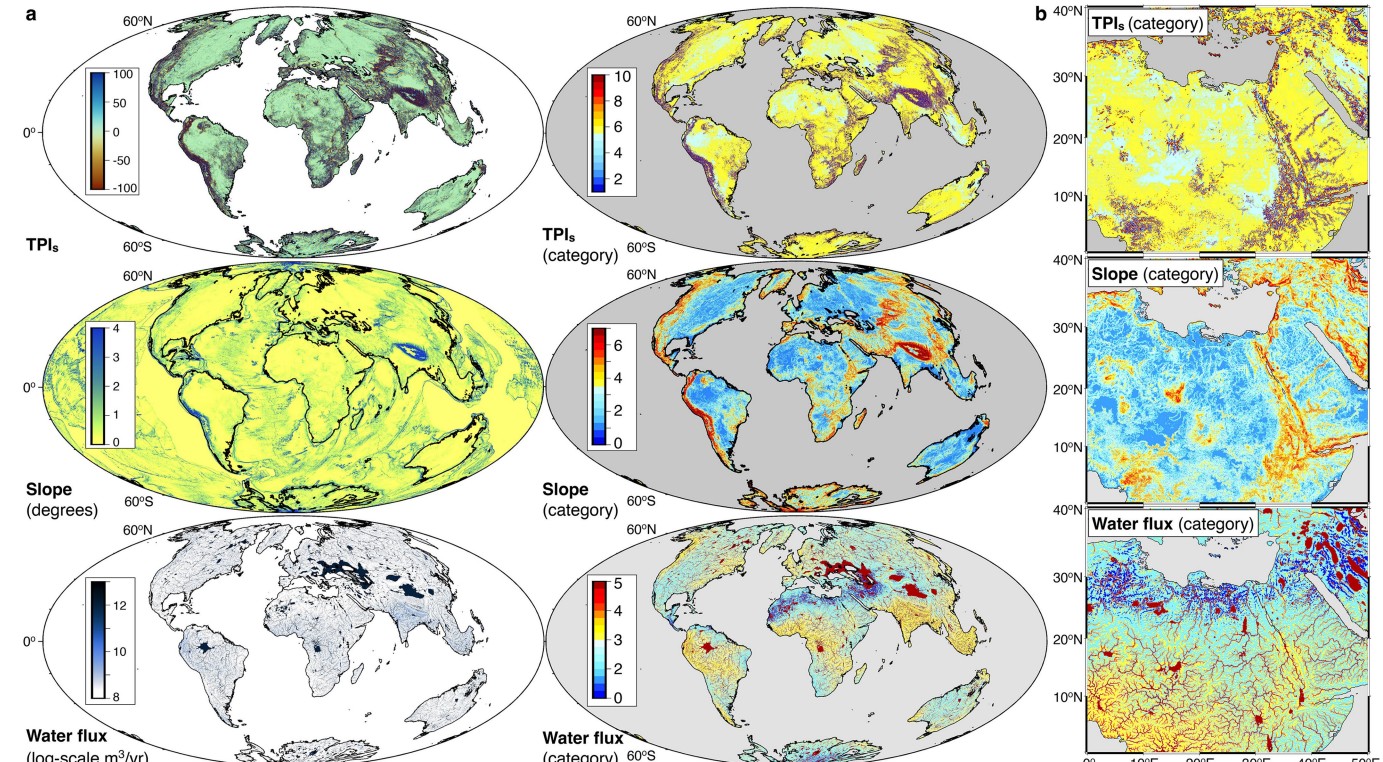

**Extended Data Fig. 9 | Hierarchical classification of physiographic relevant morphometrics derived from model outputs. a**. From standardized topographic position index (*TPI*ₛ), slopes calculated at 0.05° resolution, and reconstructed water flux, series of categories characterizing the physiography are produced at global scale (chosen example at 26 Ma). **b**. Zoomed-in maps of the physiographic categories for Northeast Africa.

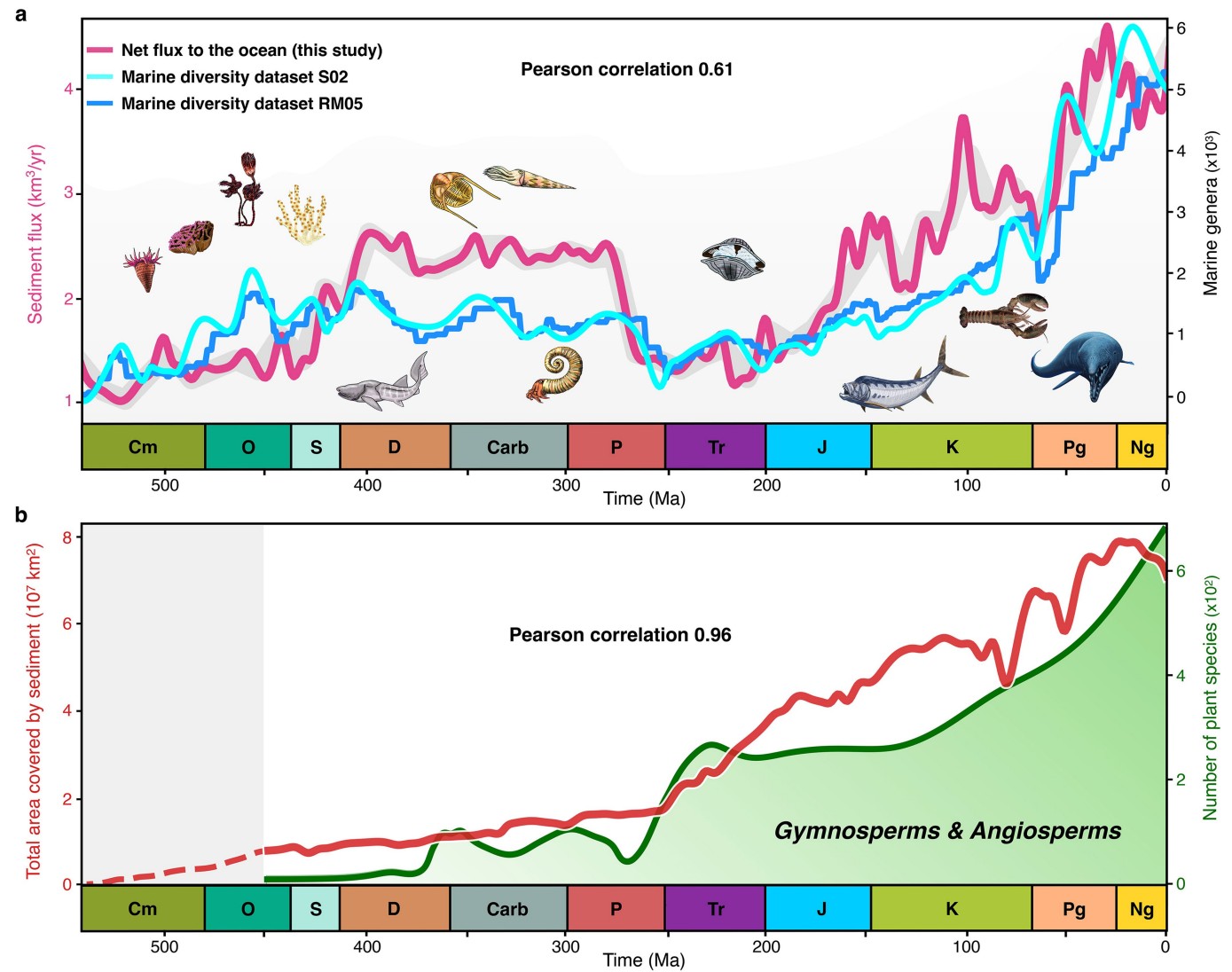

**Extended Data Fig. 10 | Biodiversity and sediment flux through time.**
**a**. Predicted trend in net sediment flux to the ocean (purple line) and estimates of total number of marine genera from S02[2] and RM05[5]; Pearson correlation of 0.61. Sepkoski's dataset (S02) downloaded from http://strata.geology.wisc.edu/jack/. Paleo-marine life by Rebecca Horwitt, available at full size and open access from https://sites.psu.edu/rhorwitt/. The grey line shows the least square regression of the sediment flux to the ocean using SciPy Savitzky-Golay filter. **b**. Cumulative total area covered by sediments preserved over time (red line) and cumulative number of gymnosperm and angiosperm species[1] (green line) showing a strongly positive correlation (Pearson of 0.96).

# Reporting Summary

## Statistics

For all statistical analyses, confirm that the following items are present in the figure legend, table legend, main text, or Methods section.

| n/a | Confirmed | |
|---|---|---|
| ☐ | ☒ | The exact sample size (*n*) for each experimental group/condition, given as a discrete number and unit of measurement |
| ☒ | ☐ | A statement on whether measurements were taken from distinct samples or whether the same sample was measured repeatedly |
| ☒ | ☐ | The statistical test(s) used AND whether they are one- or two-sided *Only common tests should be described solely by name; describe more complex techniques in the Methods section.* |
| ☒ | ☐ | A description of all covariates tested |
| ☒ | ☐ | A description of any assumptions or corrections, such as tests of normality and adjustment for multiple comparisons |
| ☐ | ☒ | A full description of the statistical parameters including central tendency (e.g. means) or other basic estimates (e.g. regression coefficient) AND variation (e.g. standard deviation) or associated estimates of uncertainty (e.g. confidence intervals) |
| ☒ | ☐ | For null hypothesis testing, the test statistic (e.g. *F*, *t*, *r*) with confidence intervals, effect sizes, degrees of freedom and *P* value noted *Give P values as exact values whenever suitable.* |
| ☒ | ☐ | For Bayesian analysis, information on the choice of priors and Markov chain Monte Carlo settings |
| ☒ | ☐ | For hierarchical and complex designs, identification of the appropriate level for tests and full reporting of outcomes |
| ☐ | ☒ | Estimates of effect sizes (e.g. Cohen's *d*, Pearson's *r*), indicating how they were calculated |

*Our web collection on statistics for biologists contains articles on many of the points above.*

## Software and code

Policy information about availability of computer code

| | |
|---|---|
| Data collection | The scientific software used in this study, goSPL, is available from https://github.com/Geodels/gospl and the software documentation can be found at https://gospl.readthedocs.io. |
| Data analysis | Jupyter, notebooks used for processing the data sets and model outputs that can be followed to reproduce some of the figure presented in the paper and can be accessed from https://github.com/Geodels/paleoPhysiography |

For manuscripts utilizing custom algorithms or software that are central to the research but not yet described in published literature, software must be made available to editors and reviewers. We strongly encourage code deposition in a community repository (e.g. GitHub). See the Nature Portfolio guidelines for submitting code & software for further information.

## Data

Policy information about availability of data

All manuscripts must include a data availability statement. This statement should provide the following information, where applicable:

- Accession codes, unique identifiers, or web links for publicly available datasets
- A description of any restrictions on data availability
- For clinical datasets or third party data, please ensure that the statement adheres to our policy

The PALEOMAP Paleodigital Elevation Models for the Phanerozoic from Scotese & Wright (2018) can be downloaded from https://doi.org/10.5281/zenodo.5460860 (last access: 25 March 2023). Paleo-precipitation maps from the HadCM3BL- M2.1aD model are available from the Bristol Research Initiative for the Dynamic Global

# Research involving human participants, their data, or biological material

Policy information about studies with human participants or human data. See also policy information about sex, gender (identity/presentation), and sexual orientation and race, ethnicity and racism.

| | |
|---|---|
| Reporting on sex and gender | N/A |
| Reporting on race, ethnicity, or other socially relevant groupings | N/A |
| Population characteristics | N/A |
| Recruitment | N/A |
| Ethics oversight | N/A |

Note that full information on the approval of the study protocol must also be provided in the manuscript.

# Field-specific reporting

Please select the one below that is the best fit for your research. If you are not sure, read the appropriate sections before making your selection.

☐ Life sciences     ☐ Behavioural & social sciences     ☒ Ecological, evolutionary & environmental sciences

For a reference copy of the document with all sections, see nature.com/documents/nr-reporting-summary-flat.pdf

# Ecological, evolutionary & environmental sciences study design

All studies must disclose on these points even when the disclosure is negative.

| | |
|---|---|
| Study description | N/A |
| Research sample | N/A |
| Sampling strategy | N/A |
| Data collection | N/A |
| Timing and spatial scale | N/A |
| Data exclusions | N/A |
| Reproducibility | N/A |
| Randomization | N/A |
| Blinding | N/A |

Did the study involve field work?     ☐ Yes     ☒ No

# Reporting for specific materials, systems and methods

We require information from authors about some types of materials, experimental systems and methods used in many studies. Here, indicate whether each material, system or method listed is relevant to your study. If you are not sure if a list item applies to your research, read the appropriate section before selecting a response.

## Materials & experimental systems

| n/a | Involved in the study |
|-----|----------------------|
| ☒ | Antibodies |
| ☒ | Eukaryotic cell lines |
| ☒ | Palaeontology and archaeology |
| ☒ | Animals and other organisms |
| ☒ | Clinical data |
| ☒ | Dual use research of concern |
| ☒ | Plants |

## Methods

| n/a | Involved in the study |
|-----|----------------------|
| ☒ | ChIP-seq |
| ☒ | Flow cytometry |
| ☒ | MRI-based neuroimaging |

