## [Peer Review File · Nature]

Manuscript Title: Landscape dynamics and the Phanerozoic diversification of the biosphere

Reviewer Comments & Author Rebuttals

Reviewer Reports on the Initial Version:

Referees' comments:

Referee #1 (Remarks to the Author):

Paper summary:

Salles et al. use a global landscape evolution model, guided by general circulation model simulations and paleogeographical reconstructions, to simulate sediment fluxes during the Phanerozoic. They distinguish between the flux of sediments exported to the ocean, and the flux of sediments stored on the continents. They also reconstruct the sediment cover preserved in endoreic basins. The authors compare the simulated temporal trends with biodiversity curves for marine invertebrates and land plants. The temporal correlation obtained between the sediment flux to the ocean and marine invertebrate biodiversity is used by the authors to suggest that the flux of sediments to the ocean drove long-term marine biodiversity changes by controlling nutrient availability and thus the carrying capacity of the environment. The temporal correlation obtained between land plant diversity and the flux of sediment stored on land, and even to greater extent with the sedimentary cover on land, is used to suggest that this sediment cover constrained the biodiversification of land plants. Additional analyses of landscape heterogeneity – using two metrics called ‘diversity’ and ‘variety’ – are further used to try and propose specific mechanisms that may have led to land plant diversification.

General comment:

The manuscript is mostly clearly written and it is always richly illustrated. The numerical simulations are state-of-the-art, but this aspect was previously evaluated and published¹ and I try to focus here on the implications for our understanding of the evolution of marine and terrestrial (i.e., land plant) biodiversity during the Phanerozoic, which really constitutes the core and novelty of the present manuscript. I have to say, I had a hard time evaluating this manuscript. I think that the results are very exciting, but at the same time, several points made me doubt of the robustness of the conclusions. Put shortly, correlations are strikingly good, I think that the authors did a great job at trying to demonstrate possible causal relationships for the evolution of land plants (through analyses of landscape heterogeneity notably); I'm just not really convinced by the link of causality regarding the marine biosphere (see comments below). Overall, I think that it is an important, major contribution that represents a significant advance and that will foster future studies; I support publication of the manuscript, after revisions. My suggestions below are intended to help improve the manuscript; they only concern the text and should be easily addressed by the authors.

Main comments:

1. A few points made me wonder whether the results obtained for the marine biosphere reflect causality, or just correlation:

- First, I wouldn't expect any direct relationship between biomass (that is effectively limited by nutrient availability) and biodiversity. At the local scale, for instance, upwellings are nutrient-rich, thus host a large biomass, but a poor diversity.
- Then, I am not really convinced by the explanations provided by the authors regarding the link between changes in sediment flux to the ocean and extinctions on page 4/27, lines 1422. The sediment flux curve has so many wiggles that it is not unexpected that extinctions roughly coincide with some excursions separated by some 2–15 Myrs. Conversely, many negative excursions in sediment flux do not coincide with biodiversity drops, notably during the Jurassic and Cretaceous.
- Above all, I think that other quantities can be correlated with biodiversity curves (and simulated sediment flux), such as the continental fragmentation index². I would more easily imagine a causal link between continental fragmentation (and the creation of new ecological niches) and biodiversity, than between ocean nutrient inventory and biodiversity.

2. I think the manuscript critically lacks discussion, including the comparison with previous work and possible limitations. Here is a list of points that I would suggest to include. If necessary, other parts of the MS could probably be shortened (I identified 2 paragraphs: last paragraph on page 2/27 and lines 11–22 on page 3/27; the first one proposes to validate the model but I think it was previously published and validated¹ while the second one focuses on the Sr curve and does not bring much to the demonstration, in my opinion).

- First, it is well possible (even more likely?) that the changing sedimentary flux drives the fossil preservation bias, rather than marine biodiversification. This hypothesis is not even invoked. That would be a major result as such.
- Then, previous work by Zaffos et al.², suggesting that continental fragmentation drove marine biodiversity trends, should be discussed in more detail.
- Similarly, the recent study by Cermeno et al.³, suggesting that the marine biosphere rarely reached its carrying capacity, would deserve a few words (disclosure for full transparency: I co-authored this study).
- Another hot topic is the impact of biases on our vision of the evolution of marine biodiversity; e.g., ref.⁴.
- Looking at Fig. 2a and Fig. 1c, I wonder to what extent the reduced erosion flux in the early Phanerozoic reflects our poor knowledge of topography at that time. The numerous and well-documented orogenic events in the Cenozoic seem to largely contribute to the rise in erosion rates at that time. I fully realize that the authors are facing current limitations in palaeogeographical reconstructions and do acknowledge the quality of the work (which is probably the best that can be done to date), but would be happy to see a few words discussing to what extent the simulated temporal trends may be impacted by these uncertainties in ancient continental topography (which are big).
- Similarly, in Fig. 4a, the sediment cover is 0 in the earliest Cambrian; I guess this is an artefact due to the model integration period? Or do the authors mean that the sediment cover was null before the Cambrian? What's the impact on the conclusions in any case?
- Last paragraph on page 6/27: it could also be argued that a strong flux of sediments (thus nutrients, supposedly) to the ocean may lead to ocean anoxia and thus constitute a killing mechanism.

In general, I think that more nuanced statements, and a discussion of the alternative explanations and model limitations, would be helpful and would actually add value to the MS without weakening it.

3. As previously stated, I don't really see the added value of the analysis of the Sr curve in this specific MS. I mean, this is definitely a nice piece of work, but I don't get how that can be used to validate the model as the authors state on lines 21–22 on page 3/27. My understanding is that these analyses mostly shed new light on the Sr curve, that's good but that's all.

4. Overall, I'm not a big fan of the conclusion. The first sentence is very provocative, and might be better toned down. Also, this is the first time here that nutrient availability is proposed as an important driver of diversification on land (lines 25–26: this idea is not stated explicitly before in the MS; see also line 30). Also, the processes listed on line 27 (climate and tectonics) are accounted for in the modeling framework used by the authors, and I don't get this sentence. The sentence on lines 27–29 is very vague and general and does not bring much in my opinion. On lines 32–33, the authors further state that "physiographic diversity [...] covaries with erosion rates" but I can't see that when comparing Fig. 4b (upper panel) with Fig. 2a (and actually don't necessarily understand why this would necessarily always be the case).

Minor comments:

1. Page 4/27, lines 24–26: I think that the authors mean that abiotic factors *controlled* speciation and extinction, which together drive biodiversity changes through time?
2. Page 5/27, lines 10–11: I don't get how an increase in sediment cover can contribute to rejuvenating the bare rock? I would expect this cover to make the system tend towards the transport-limited erosional regime?
3. Page 6/27, line 9: "along with a decrease in continental deposition flux": I thought the authors previously stated that the sediment cover, rather than the flux, was key? Please clarify this point.
4. Fig. 3: please define the white lines in the caption.
5. Fig. 4c: what is SedCover, and what is $0.019 + 0.27Pvar + 0.61 SecCover$?

Technical comments:

- Page 4/27, line 6: I guess 'its' refers to the net sediment flux; please reword to clarify.
- Page 4/27, line 7: Similarly, please clarify what 'it' refers to.

References cited: 1. Salles, T. et al. Hundred million years of landscape dynamics from catchment to global scale. *Science* 379, 918–923 (2023). 2. Zaffos, A., Finnegan, S. & Peters, S. E. Plate tectonic regulation of global marine animal diversity. *Proceedings of the National Academy of Sciences* 114, 5653–5658 (2017). 3. Cermeño, P. et al. Post-extinction recovery of the Phanerozoic oceans and biodiversity hotspots. *Nature* 607, 507–511 (2022). 4. Harper, D. A. T., Cascales-Miñana, B., Kroeck, D. M. & Servais, T. The palaeogeographical impact on the biodiversity of marine faunas during the Ordovician radiations. *Global and Planetary Change* 207, 103665 (2021). 5. Lenton, T. M., Crouch, M., Johnson, M., Pires, N. & Dolan, L. First plants cooled the Ordovician. *Nature Geoscience* 5, 86–89 (2012).

Referee #2 (Remarks to the Author):

Review of “Physiography dynamics and the Phanerozoic diversification of the biosphere” by Salles et al.

In this paper, the authors explore the relationship between tectonic/precipitation/landscape change and marine/continental biodiversity change, over the Phanerozoic. They make use of existing floral/faunal datasets to quantify global metrics of biodiversity, and compare these with estimates of sediment fluxes and sediment area predicted by an erosion/landscape model.

Ultimately, they find a good correlation between model-predicted net sediment flux to the ocean and observed marine biodiversity, and between model-predicted continental sediment area and observed continental biodiversity, and they hypothesise that these are linked causally.

Overall, I think that this paper is very interesting and important, and potentially publishable. However, the results are currently presented without any assessment of uncertainty, in particular in terms of the spatial fields driving the results, i.e. the paleogeography, and the climate model-derived precipitation, which makes the predictions almost impossible to interpret, in my view. Sensitivity studies to the uncertain inputs to the model are required in order to be able to assess the robustness of the results.

Major comments:

The paper needs to be much more critical of the climatic inputs. The precipitation from the model is taken as “truth”; however, the precipitation fields from Valdes et al (2021) are associated with high levels of uncertainty, which vary in space and time. e.g. the discussion around lines 21-28 discusses some global precipitation changes, but these will be highly sensitive to (a) the climate model used, internal parameters in the climate model, the CO₂ assumed in the climate model, and indeed the paleogeography in the climate model. These uncertainties need to be acknowledged, explained, and their impact on the results assessed (e.g. through sensitivity studies with different climate precipitation inputs). Many sensitivity studies with this climate model exist (published and unpublished) which should be used here. In addition, simulations with other models exist (e.g. Li et al, 2022) over these timescales, which will have significantly different precipitation patterns, and these could also be used in order to characterise this uncertainty.

I also have some concerns about the use of precipitation in the erosion model. The modelled precipitation represents the total water downward flux at the surface, and yet much of this is evaporated very quickly, and does not play a role in erosion (at least in terms of riverine erosion). I would have thought that the net water flux, P-E to be the relevant variable here, as this is the precipitation that is available to drive erosion. These variables should all be available from the climate model. And what about seasonality in precipitation – due to erosion being a non-linear function of precipitation, I would expect the seasonal precipitation to be required, in particular in monsoon regions. Again, these variables are available from the climate model. Finally, it is stated that (methods, page 1, line 17) “In our formulation, the weathering impact of precipitation....”.

However, weathering is also highly dependent on local temperature, and yet local temperature is not used as input to the erosion/weathering model, as far as I can tell (and yet again is available for these climate model simulations).

Specific comments:

Page 1, lines 23-37: The possible impact of sampling bias on the apparent “monotonic increase of diversity over time in the terrestrial realm” should be acknowledged and discussed in the introduction. Especially for the more recent Phanerozoic, e.g. how confident are we that e.g. the Last Glacial Maximum was more diverse than the mid Cretaceous?

Page 1, line 39: “The changing paleo-geography in turns modulates the atmosphere”. I do not know what “modulates the atmosphere” means. This is very unclear. A standard scientific term should be used. In addition, specific examples would help, e.g. topographic highs generating Rossby waves which influence large-scale atmospheric circulation and therefore heat and moisture transport and ultimately temperature and precipitation patterns. In addition, just as important as the atmosphere is the ocean. Paleogeographic changes affect ocean gateways and basins and therefore ocean circulation and heat transport, which ultimately can affect continental climate. There are large bodies of seminal, plus more recent, literature on this that should be cited.

Page 3. Some checking is done of the modern results of the model compared with modern observations and of the self-consistency of the modelling framework (e.g. p3, starting line 31), and of the paleo fluxes (p3, lines 11-22), but the modelled paleo precipitations, which are of crucial importance here, are not directly validated. Again, there are several papers that have evaluated the precipitation in this climate model (and others) for various past time periods. However, the paper needs to acknowledge that modelled precipitation is highly uncertain. Model predictions vary hugely just for 100 years in the future (see e.g. IPCC AR6), let alone for 500 million years ago in the past, so this should be acknowledged and discussed.

Page 4, lines 5-12: Correlation is found between sediment flux and marine organisms. There is a plausible and previously-suggested mechanism for this, but still, correlation does not imply causation. This becomes even more apparent for the terrestrial, where a correlation is found between available sediment and continental biodiversity, but this seems rather tenuous; there are many other aspects that could be playing an important role, and just because variable A and variable B increase through the Phanerozoic doesn't mean that A causes B. This should be acknowledged in the text. Even better, this is where an actual biodiversity model (e.g. that of Erin Saupe) could be used to support these ideas.

Figure 3,4: What causes this variability in modelled sediment flux and sediment area? Is it primarily coming from the time-varying modelled precipitation, or from the time-varying tectonic change, or both? Sensitivity studies in which various aspects (e.g. precipitation) are held constant in time would enable this question to be answered.

P5, line 13: “increasing physiographic variety”. How robust is this, given the huge uncertainties in the reconstructed paleogeographies, especially prior to the Cretaceous where the paleogeographies are

even more uncertain.

What about the fact that biological evolution and biodiversity themselves cause changes in erosion rates, driven by factors such as the development of root systems, or even the evolution and spread of earthworms. This paper views the causal process very much one directional, whereas in reality the biotic and sedimentological factors are tightly coupled.

P6, line 26: “climate control on hydrological and CO₂ conditions” – this is a very odd thing to say here, because the precipitation changes (which themselves are driven by CO₂ changes) are determining the predicted sediment and nutrient fluxes.

Methods: Global physiography model. What spatial (and temporal) resolution is the goSPL model run at? I assume the same resolution as the paleogeographic / paleoclimatic input data, but this is not made explicit. Actually, I see that this is mentioned in the next section, but it should be mentioned here first.

Methods: Paleo-elevation and paleo-precipitation forcing. For the paleogeography inputs, I agree with the authors that the paleogeographies are uncertain – this is particularly true in the earlier Phanerozoic. It is mentioned that other paleogeographies could be used. Here, I would like to see some quantification of the uncertainty that this would bring to the results (e.g. by using an alternative reconstruction for one or two of the timeslices (e.g. Straume et al, 2020)).

Methods: Paleo-elevation and paleo-precipitation forcing. There are several issues here. First, there is a complete disconnect between the resolution of the input precipitation fields (3.75 degree by 2.5 degrees, i.e. about 270km at the equator) and the resolution that this is being interpolated to (5km globally!). In reality, precipitation patterns have a huge amount of spatial variability which is lost due to the low resolution nature of the climate model. Due to the highly non-linear nature of precipitation-driven erosion and weathering, spatial downscaling of the climate model outputs should be applied (related e.g. to the high resolution topography, and e.g. moisture transport direction). Secondly, as commented above, the modelled precipitation is highly uncertain, and sensitivity studies should be carried out to quantify the impacts of this uncertainty on the results.

Minor comments:

Line 38: “dance of continents” is somewhat informal. There are many such instances of use of informal/vague language, e.g. “burst in precipitations”, “booming flux of sediments”.

Figure 1: Panel (a) looks pretty, but misses out half the planet. For a scientific paper, this is better represented as a map. Also, you need to justify in the caption/text why the specific regions in panel (b) are chosen.

Many terms are used to describe the driving forces of diversification, and to me the difference between them (if any) is unclear. Through the paper, these terms need to be defined and used consistently. For example, in the Abstract alone, the following are all used: “geological or climatic... changes in the physical environment”, “geodynamic and climatic forcing”, “Earth’s physiography”, “landscape dynamics”.

It would be very informative to see maps in the Supp Info of the input precipitation from the climate model (especially if this is downscaled to a higher resolution than the input climate model precipitation, see comments above).

References

Li, X., Hu, Y., Guo, J. et al. A high-resolution climate simulation dataset for the past 540 million years. *Sci Data* 9, 371 (2022). <https://doi.org/10.1038/s41597-022-01490-4>.

Straume, E. O., Gaina, C., Medvedev, S., and Nisancioglu, K. H., 2020, Global Cenozoic Paleobathymetry with a focus on the Northern Hemisphere Oceanic Gateways: *Gondwana Research*, v. 86, p. 126-143.

Valdes, P. J., Scotese, C. R., and Lunt, D. J.: Deep ocean temperatures through time, *Clim. Past*, 17, 14831506, <https://doi.org/10.5194/cp-17-1483-2021>, 2021.

POINT-BY-POINT RESPONSES TO REVIEWERS

We were very pleased with the outcomes of the review process, both reviewers acknowledged the novelty and significance of our work. They salute the quality of the illustrations, and point that “*it is an important, major contribution that represents a significant advance and that will foster future studies*”. As a result, they both recommended publication after revisions pending few suggestions and comments (Reviewer 1: “*I support publication of the manuscript, after revisions. My suggestions below are intended to help improve the manuscript; they only concern the text and should be easily addressed by the authors*” - Reviewer 2: “*I think that this paper is very interesting and important, and potentially publishable*”) that we address in this document.

We would like to thank the reviewers for their thoughtful comments and efforts towards improving our manuscript. In the following, we first answer high-level suggestions of each reviewer (major comments) and the modifications that were made to address them. We then respond to each specific comment (minor and technical).

REVIEWER 1

Major comments

1. First, I wouldn't expect any direct relationship between biomass (that is effectively limited by nutrient availability) and biodiversity. At the local scale, for instance, upwellings are nutrient rich, thus host a large biomass, but a poor diversity.

As mentioned by the reviewer, the upper limits of biomass production are clearly determined by the abiotic environment (e.g., nutrient availability), but the relationships between biomass and biodiversity have been debated for decades. A full review would be inappropriate in the current article, and we only point to the most salient aspects of the debate, on which our hypothesis is based.

The general view is that increasing primary production first increases diversity, which then starts to decrease after a certain point, as communities with high biomass are usually dominated by highly productive competitive species (Mittelbach et al., 2001). Reported trends in field studies (Wardle et al., 1997; Luo et al., 2019, to cite a few) suggest different evolutions from positive or negative patterns in which biodiversity increases or decreases linearly with productivity to unimodal (hump-shaped) patterns in which biodiversity peaks at intermediate productivity (Craven et al., 2016). The work from Duffy et al. (2017), using >60 field studies, shows a positive association (i.e., higher biodiversity and higher biomass production).

Here and following Gross & Cardinale (2007) and Cardinale et al. (2009), we speculate that community biomass and diversity are both influenced by the resource supply rate of the habitat. We have added references in the corresponding sentence in the manuscript (line 108/109 p. 3) to clarify the proposed relation.

References

Cardinale, B. J. et al. Does productivity drive diversity or vice versa? A test of the multivariate productivity–diversity hypothesis in streams. *Ecology* **90**, 1227–1241 (2009).

Gross, K. & Cardinale, B. J. Does species richness drive community production or vice versa? Reconciling historical and contemporary paradigms in competitive communities. *Amer. Nat.* **170**, 207–220 (2007).

Craven, D. et al. Plant diversity effects on grassland productivity are robust to both nutrient enrichment and drought. *Philos. Trans. R. Soc. Lond. B Biol. Sci.* **371**, 20150277 (2016).

Duffy, J. E. et al. Biodiversity effects in the wild are common and as strong as key drivers of productivity. *Nature* **549**, 261–264 (2017).

Luo, W. et al. Parameterization of biodiversity–productivity relationship and its scale dependency using georeferenced tree–level data. *J. Ecol.* **107**, 1106–1119 (2019).

Mittelbach, G. G. et al. What is the observed relationship between species richness and productivity? *Ecology* **82**, 2381–2396 (2001).

Wardle, D. A. et al. The influence of island area on ecosystem properties. *Science* **277**, 1296–1299 (1997).

2. I am not really convinced by the explanations provided by the authors regarding the link between changes in sediment flux to the ocean and extinctions. The sediment flux curve has so many wiggles that it is not unexpected that extinctions roughly coincide with some excursions separated by some 2–15 Myrs. Conversely, many negative excursions in sediment flux do not coincide with biodiversity drops, notably during the Jurassic and Cretaceous.

Following reviewer’s comment, we have modified the corresponding section (lines 131-147 p. 3/4) on the link between sediment flux and extinctions, because indeed the number of wiggles in predicted net sediment flux to the ocean requires cautiousness.

However, we also expand on our initial explanation and suggest that sediment flux deprivation could represent a continuous disturbance favourable to extinction in a sustained manner. As a result, we highlight that like climate, volcanism and sea level change, sediment flux could be considered in the press-pulse framework when evaluating the compounding effects which triggered episodes of extinctions in Phanerozoic history (Arens and West, 2008).

Reference

Arens, N. C. & West, I. D. Press-pulse: a general theory of mass extinction? *Paleobiology* 34, 4, 456–471 (2008).

3. I think that other quantities can be correlated with biodiversity curves (and simulated sediment flux), such as the continental fragmentation index. I would more easily imagine a causal link between continental fragmentation (and the creation of new ecological niches) and biodiversity, than between ocean nutrient inventory and biodiversity.

Figure R1. Plate tectonics and the Phanerozoic marine biodiversity. a. The index of continental block fragmentation derived from Zaffos et al. (2017) (orange curve) with value close to 1 indicating no plates are touching and a value of zero corresponding to contiguous continental blocks arranged in a single mass. Pearson correlation (0.46) indicates a positive but moderate relationship with the number of marine families (black line from Sepkoski’s dataset (Sepkoski, 2002)). b. Correlations between the fragmentation index and the total number of marine genera from Sepkoski (2002) and Rohde & Muller (2005) show a slightly stronger but still moderate positive trends (Pearson values of 0.58 and 0.54 respectively).

Based on reviewer's suggestions, we show in Fig. R1 the evolution of the continental fragmentation index from Zaffos et al. (2017) and analyse its relationship with Phanerozoic marine diversity.

As mentioned by the reviewer, continental fragmentation has been proposed as one of the critical drivers of marine biodiversity increase. It is specially described for specific periods of Earth history such as during the fragmentation of Rodinia in the Cambrian-Ordovician or in the Late Mesozoic with the breakup of Pangaea (Brasier & Lindsay, 2001; Miller & Mao, 1995; Owen & Crame, 2002; Zaffos et al., 2017). Whilst true in those cases, Fig. R1 shows that the trajectory of Phanerozoic marine diversity (both families and genera, available from the paleo-record) exhibits a moderate correlation with continental fragmentation (exemplified by calculated Pearson coefficients).

On the contrary, the correlation based on predicted net sediment flux to the ocean (Fig. 3 in the manuscript) shows a strong positive relationship (Pearson close to 0.9, quoting the reviewer "correlations are strikingly good"). In our views, our approach accounts for the compounding effects of tectonics, climates, and landscapes. Tectonically driven shifts in paleogeography (creation and destruction of geological barriers) and global ocean-atmospheric circulation ultimately impact sediment transport. By combining these forcings, calculated mean sediment distribution for the Phanerozoic landscape shows a remarkable congruence with the diversification of marine life.

Following reviewer's comment, we have broadened the scope of the drivers (mainly focussed on ocean nutrient inventory in the initial manuscript) and modified accordingly the section on marine diversification and net sediment flux to the ocean (line 156-160 p. 4).

References

Brasier, M.D. & Lindsay, J.F. Did supercontinental amalgamation trigger the "Cambrian Explosion". *The Ecology of the Cambrian Radiation* (Columbia Univ Press, New York), 69–89 (2001).

Miller, A.I. & Mao, S. Association of orogenic activity with the Ordovician radiation of marine life. *Geology*, **23**, 305–308 (1995).

Owen, A. & Crame J. Palaeobiogeography and the Ordovician and Mesozoic-Cenozoic biotic radiations. *Geol Soc Spec Publ.*, **194**, 1–11 (2002).

Rohde, R. A. & Muller, R. A. Cycles in fossil diversity. *Nature* **434**, 208–210, (2005).

Sepkoski, J. J. A compendium of fossil marine animal genera. *Bull. Am. Paleontol.* **363**, 1–560 (2002).

Zaffos, A. et al. Plate tectonic regulation of global marine animal diversity. *Proceedings of the National Academy of Sciences* **114**, 5653–5658 (2017).

4. I identified 2 paragraphs: last paragraph on page 2/27 and lines 11–22 on page 3/27; the first one proposes to validate the model, but I think it was previously published and validated while the second one focuses on the Sr curve and does not bring much to the demonstration, in my opinion.

The method proposed in the paper slightly differs from our previously published Science paper (Salles et al., 2023), but we agree with the reviewer and have moved the validation section using the Sr seawater curve comparison to the Methods.

Reference

Salles T. et al. Hundred million years of landscape dynamics from catchment to global scale. *Science*, **379**, 918–923 (2023).

5. It is well possible (even more likely?) that the changing sedimentary flux drives the fossil preservation bias, rather than marine biodiversification. This hypothesis is not even invoked. That would be a major result as such.

We agree with the reviewer, the strong relationship between the marine biodiversification and our predicted evolution of sedimentary flux could actually illustrate fossil preservation bias. This is even more likely for the marine palaeobiological record because the marine environment is generally less affected by erosional processes when compared to terrestrial regions. This relates with several other studies (Alroy et al., 2001) that already discussed the incompleteness and spatial heterogeneity of the marine fossil record.

Following the reviewer hypothesis, we are now discussing this potential bias in the relevant section and added the following lines 148-156 in p. 4.

Reference

Alroy, J. et al. Effects of sampling standardization on estimates of Phanerozoic marine diversification. *Proc. Natl. Acad. Sci.*, **98**, 6261–6266 (2001).

6. Previous work by Zaffos et al. (2017), suggesting that continental fragmentation drove marine biodiversity trends, should be discussed in more detail.

As explained in our response to the previous point raised by the reviewer (point 3), we have now broadened the scope of the drivers in the corresponding section on marine diversification.

7. The recent study by Cermeño et al. (2022), suggesting that the marine biosphere rarely reached its carrying capacity, would deserve a few words.

We agree and refer to this specific paper by Cermeño et al. (2022) in the last paragraph of the marine biodiversification section. Our model supports a similar idea where abiotic controls might be playing a role as equally determinant as speciation and extinction rates over geological time scale.

Reference

Cermeño, P. et al. Post-extinction recovery of the Phanerozoic oceans and biodiversity hotspots. *Nature* **607**, 507–511 (2022).

8. Another hot topic is the impact of biases on our vision of the evolution of marine biodiversity (e.g., Harper et al. (2021)).

We surely agree but remain unsure about the biases that the reviewer is referring to here, for example we could identify several preservation / taphonomy / exposure biases. Harper et al. (2021) highlights a strong regional bias for the Ordovician diversity, with specific faunas dominating the currently available global databases. In the revised version of the manuscript, we are now discussing the preservation bias (as explained in our response to point 5).

Reference

Harper, D. A. T. et al. The palaeogeographical impact on the biodiversity of marine faunas during the Ordovician radiations. *Global and Planetary Change* **207**, 103665 (2021).

9. Looking at Fig. 2a and Fig. 1c, I wonder to what extent the reduced erosion flux in the early Phanerozoic reflects our poor knowledge of topography at that time. The numerous and well-documented orogenic events in the Cenozoic seem to largely contribute to the rise in erosion rates at that time. I fully realize that the authors are facing current limitations in palaeogeographical reconstructions and do acknowledge the quality of the work (which is probably the best that can be done to date) but would be happy to see a few words discussing to what extent the simulated temporal trends may be impacted by these uncertainties in ancient continental topography (which are big).

We thank the reviewer for acknowledging the *quality of the work* and the limitations we are facing due to some of the uncertainties related (but not limited to) the representation of palaeogeographical reconstructions. This aspect related to paleo-elevation uncertainties is discussed and illustrated in our previous paper (Salles et al., 2023).

Following reviewer's comment, we are discussing how those issues could affect predicted temporal trends (Methods line 158-230 p. 4-6); in addition, we now also dedicate a full chapter (Methods) to the sensitivity analysis of the impact of paleo-climate models following reviewer 2 comments.

Reference

Salles T. et al. Hundred million years of landscape dynamics from catchment to global scale. *Science*, **379**, 918–923 (2023).

10. Similarly, in Fig. 4a, the sediment cover is 0 in the earliest Cambrian; I guess this is an artefact due to the model integration period? Or do the authors mean that the sediment cover was null before the Cambrian? What's the impact on the conclusions in any case?

As pointed out by the reviewer, the cumulative curve in Fig. 4a starts at 0 at the beginning of our simulation (540 Ma). This is indeed due to the model integration period. There is no reason to believe that sediment cover was null before the Cambrian. However as shown in Fig. 2a, continental sediment flux remains low for a long period of time (for more than 100 Myr - up to the Carboniferous period). As a result, we hypothesize that sediment cover over this period would have had a moderate impact on terrestrial biodiversification. Even in the case of a large pre-Phanerozoic sediment cover, those soft sediments would have likely turned into bare hard rocks by 450 Ma (during the Silurian when early vascular plants started diversifying). More importantly, we state that sediment cover is a necessary but non-unique condition for the development of life. Only after the inception of life does sediment cover set the carrying capacity.

In the revised version of the paper, we are now discussing further these hypotheses emphasizing on the idea that for soil and sediment cover to have an effect of diversification, we need life to co-evolve (line 179-181, 193 p. 4).

11. Last paragraph on page 6/27: it could also be argued that a strong flux of sediments (thus nutrients, supposedly) to the ocean may lead to ocean anoxia and thus constitute a killing mechanism.

We agree with the reviewer's comment, and this is indeed something we investigated as it could be the case.

Figure R2. Relationships between ocean sediment flux, oceanic anoxic events, and marine biodiversity. Evolution of predicted net sediment flux delivered to the ocean (purple), the number of marine families (black line from Sepkoski's dataset (Sepkoski, 2002)), and the timing of major oceanic anoxic events (OAEs) during the Mesozoic and Cenozoic (blue vertical lines (Jenkyns, 2010)). Possible correspondences between specific OAEs and the predicted sediment flux to the ocean are highlighted and include those of late Hauterivian (Faraoni event, ~130 Ma), early Albian (Paquier event, OAE 1b, ~111 Ma), late Santonian (OAE 3, ~85 Ma), and The Paleocene–Eocene Thermal Maximum (PETM, ~56 Ma).

In Fig. R2, we show the distribution of oceanic anoxic events (OAEs) with respect to predicted sediment flux since the Mesozoic. While some of these events are concomitant with predicted sediment pulses, the relationship is not straightforward. Noteworthy, the relationship between anoxic events and extinctions is also not clear (Ernst and Youbi, 2017).

As suggested by the reviewer, accelerated hydrological cycle, increased continental weathering, and enhanced nutrient discharge are considered as major drivers behind OAEs. Full-scale OAE would take *only* thousands of

years to develop (Watson, 2016) and one could argue that the temporal resolution of our study (5 Myr time-slice) is too coarse to capture its effect. Similarly, the evolution of the number of marine families from Sepkoski (2002) lacks the temporal and spatial resolution required to evaluate OAEs impact on marine diversity.

One important aspect of OAE development relates to sudden increase in phosphorus supply (phosphate is considered as the main limiting nutrient favouring hypoxic ocean condition). Large igneous provinces (LIPs) contain substantial phosphorus, are highly susceptible to chemical weathering, and prominent OAEs occurred at the same time or very shortly after their eruptions (Jenkyns, 2010). As pointed out in our paper, a possible avenue to refine our prediction would be to account for the “*variable lithologies of eroded continental rocks over time and space*”. As an example, by mapping LIPs distribution over space and time, our approach could be adapted to impose maps of fast-weathering rocks and track where these rocks are distributed. This could then be used to quantify the supply and estimate the composition of nutrients to the oceans.

We have modified the revised manuscript and added in the discussion some of these ideas (line 138-143 p. 4).

References

Richard, E. & Youbi, N. How Large Igneous Provinces affect global climate, sometimes cause mass extinctions, and represent natural markers in the geological record, *Palaeogeog., Palaeoclim., Palaeoeco.*, **478**, 30-52 (2017).

Jenkyns, H. C. Geochemistry of oceanic anoxic events, *Geochem. Geophys. Geosyst.*, **11**, Q03004 (2010).

Sepkoski, J. J. A compendium of fossil marine animal genera. *Bull. Am. Paleontol.* **363**, 1-560 (2002).

Watson, A. J. Oceans on the edge of anoxia. *Science*, **354**, 1529-1530 (2016).

12. In general, I think that more nuanced statements, and a discussion of the alternative explanations and model limitations, would be helpful and would add value to the MS without weakening it.

We have made several changes in the manuscript with added explanations (based on the Reviewer previous suggestions) and added a thorough discussion on model limitations in the Methods section (p. 4 to 6).

13. As previously stated, I don't really see the added value of the analysis of the Sr curve in this specific MS. I mean, this is a nice piece of work, but I don't get how that can be used to validate the model as the authors state on lines 21–22 on page 3/27. My understanding is that these analyses mostly shed new light on the Sr curve, that's good but that's all.

Following reviewer's suggestion, and as stated above, we have moved the analysis of the Sr curve to the Method section.

14. Overall, I'm not a big fan of the conclusion. The first sentence is very provocative and might be better toned down.

Following reviewer's comment, we have modified the sentence.

15. The processes listed on line 27 (climate and tectonics) are accounted for in the modelling framework used by the authors, and I don't get this sentence.

Following reviewer's comment, we have modified the sentence.

16. The sentence on lines 27–29 is very vague and general and does not bring much in my opinion.

As suggested by the reviewer, we have removed this sentence in the revised version.

17. On lines 32–33, the authors further state that “physiographic diversity [...] covaries with erosion rates” but I can't see that when comparing Fig. 4b (upper panel) with Fig. 2a (and don't necessarily understand why this would necessarily always be the case).

Following reviewer's comment, we have deleted this statement in the revised version.

Minor/Technical comments

1. Page 4/27, lines 24–26: I think that the authors mean that abiotic factors *controlled* speciation and extinction, which together drive biodiversity changes through time?

We have rephrased the sentence following reviewer's suggestion.

2. Page 5/27, lines 10–11: I don't get how an increase in sediment cover can contribute to rejuvenating the bare rock? I would expect this cover to make the system tend towards the transport-limited erosional regime?

This is true, our wording was misleading, and we have changed "rejuvenating" with "replacing".

3. Page 6/27, line 9: "along with a decrease in continental deposition flux": I thought the authors previously stated that the sediment cover, rather than the flux, was key? Please clarify this point.

We observe an initial rise in Gymnosperm diversity during the early phases of Pangea breakup, concomitant with an increase in sediment cover, an increase in continental deposition flux, and an increased in physiographic variety (Fig. 2a & 4). Then diversity plateaued between the Jurassic and Cretaceous periods.

We attribute this period of stabilisation to the decrease in both continental deposition flux and physiography diversity as well as a relatively steady sediment coverage.

We have modified this section in the revised manuscript.

4. Fig. 3: please define the white lines in the caption.

White lines mark the transition between the Cambrian, Palaeozoic, and Modern marine families. We have added the definition to the caption in Fig. 3.

5. Fig. 4c: what is SedCover, and what is '0.019 + 0.27Pvar +0.61 SedCover'?

In the figure caption we define *SedCover* as the cumulative area covered by sediments.

To estimate the relationship between the 2 independent quantitative variables *SedCover* and *PVAR* and the diversity of vascular plant, we performed a multivariate regression analysis (Ordinary Least Squares regression (OLS)) that calculates the coefficients of the linear regression equation ($0.019 + 0.27Pvar + 0.61 SedCover$).

Based on reviewer's point, we have changed Fig. 4c and updated the caption accordingly to remove any confusion.

6. Page 4/27, line 6: I guess 'its' refers to the net sediment flux; please reword to clarify. Similarly, please clarify what 'it' refers to on line 7.

The 'its' and 'it' refer to the reconstructed net sediment flux and we have modified the text for clarity in the revised version.

REVIEWER 2

Major comments

1. The paper needs to be much more critical of the climatic inputs. The precipitation from the model is taken as “truth”; however, the precipitation fields from Valdes et al (2021) are associated with high levels of uncertainty, which vary in space and time. e.g., the discussion around lines 21-28 discusses some global precipitation changes, but these will be highly sensitive to (a) the climate model used, internal parameters in the climate model, the CO₂ assumed in the climate model, and indeed the paleogeography in the climate model. These uncertainties need to be acknowledged, explained, and their impact on the results assessed (e.g., through sensitivity studies with different climate precipitation inputs).

Valdes et al. (2021) conducted 109 time-slice simulations that cover the entire Phanerozoic, using a coupled atmosphere–ocean–vegetation model at a resolution of 3.75°×2.5°. We agree with the reviewer’s comment, the paleo-precipitations from Valdes et al. (2021) come with uncertainties and we do not *per-se* consider them as the “truth”.

Estimating the sensitivity of the results to the paleo-climatic reconstructions is part of the set of new simulations that we are presenting in the revised version of the manuscript (more about this in the responses to the following comments). We have also modified the text to acknowledge and explain some of the limitations of these simulations in the Method section (p. 4 to 6).

More importantly, the approach that is described in our study is not tangled to a specific reconstruction (either climatic, tectonic, or topographic) and other ones could certainly be used (as discussed in Salles et al., 2023). However, most published paleo-climate reconstructions tackle one time period, and maybe use one or a handful of model simulations. In addition, it is preferable, for global consistency, to use paleo-climatic reconstructions that are tied to the same tectonic reconstructions that we use in our simulations, which further restrain the possible choices. Because of the temporal range provided by Valdes et al. (2021) (entire Phanerozoic) as well as the relatively high resolution (0.1°) of the paleo-geographic reconstructions from Scotese & Wright (2018), we believe that these two datasets are the best published and publicly available options to date (as also emphasized by reviewer #1).

References

Salles T. *et al.* Hundred million years of landscape dynamics from catchment to global scale. *Science*, **379**, 918–923 (2023).

Scotese, C. & Wright, N. Paleo digital Elevation Models (PaleoDEMS) for the Phanerozoic. **PALEOMAP Project** (2018).

Valdes, P. *et al.* Deep ocean temperatures through time. *Clim. Past.* **17**, 1483–1506 (2021).

2. Many sensitivity studies with this climate model exist (published and unpublished) which should be used here. In addition, simulations with other models exist (e.g., Li et al, 2022) over these timescales, which will have significantly different precipitation patterns, and these could also be used to characterise this uncertainty.

To our knowledge, there have been few simulation studies other than the one from Valdes et al. (2021) covering the whole Phanerozoic Eon.

- Landwehrs et al. (2021), performed 40 time-slice simulations for the period from 255 million years ago (Ma) to 60 Ma, using the CLIMBER-3 α Earth System Model of Intermediate Complexity (EMIC) that has a relatively coarse spatial resolution.
- The Bristol Research Initiative for the Dynamic Global Environment (BRIDGE) group at University of Bristol has produced large datasets of paleoclimate simulations (Lunt et al., 2016; Farnsworth et al, 2019).
- And more recently the one from Li et al. (2022) that uses the Community Earth System Model version 1.2.2 (CESM1.2.2), a coupled climate model that includes atmosphere, ocean, land, sea-ice and river components. This last model has a higher resolution (~1°) than the one from Valdes et al. (2021) but comes with half the number of time-slices.

Based on our reviewer's suggestion, we have used the newly published climate dataset from Li et al. (2022) and are presenting the impact of using this higher resolution precipitation paleo-maps on the predicted sediment flux. We agree that this second set of simulations is important to test our hypothesis. The major outcome is that, in spite of the different climatic conditions, our estimated denudation rates and sediment fluxes (both oceanic and terrestrial) remain almost similar (albeit slightly higher with the model from Li et al.) (Extended Data Fig. 6). This new simulation thus nicely corroborates our initial interpretations.

Noteworthy, a technical limitation arises from the computational cost. The simulations we performed for this study represent a huge amount of CPU time (worth approximately 10 years of CPU time, including 3.5 years CPU time, for the complementary simulation presented in the revised version). We therefore have had to amend our text to highlight the limitations of the work rather than carry out other additional simulations than the two presented in the new version of the paper (Valdes et al., 2021; and Li et al., 2022) (Methods - line 204-230 p. 5/6).

References

Landwehrs, J. *et al.* Investigating Mesozoic climate trends and sensitivities with a large ensemble of climate model simulations. *Paleoceanogr. Paleoclimatol.* **36**, e2020PA004134, (2021).

Li, X. *et al.* A high-resolution climate simulation dataset for the past 540 million years. *Sci Data* **9**, 371 (2022).

Lunt, D. J. *et al.* Palaeogeographic controls on climate and proxy interpretation. *Clim. Past* **12**, 1181–1198 (2016).

Farnsworth, A. *et al.* Climate sensitivity on geological timescales controlled by nonlinear feedbacks and ocean circulation. *Geophys. Res. Lett.* **46**, 9880–9889 (2019).

Valdes, P. *et al.* Deep ocean temperatures through time. *Clim. Past.* **17**, 1483–1506 (2021).

3. I also have some concerns about the use of precipitation in the erosion model. The modelled precipitation represents the total water downward flux at the surface, and yet much of this is evaporated very quickly, and does not play a role in erosion (at least in terms of riverine erosion). I would have thought that the net water flux, P-E to be the relevant variable here, as this is the precipitation that is available to drive erosion. These variables should all be available from the climate model.

Indeed, the paleo-evaporation maps are available from the climate model (Valdes et al., 2021).

In fact, we did exactly what the reviewer suggested and followed the same approach as in our previous paper (Salles et al., 2023). Because goSPL does not account for evaporation, we extracted the paleo-precipitation minus paleo-evaporation maps for the relevant time steps and used these maps to compute the net water flux (runoff) (PA in Eq. 1). We have added it in the method section (Methods line 18 p. 1).

It is worth noting that here the runoff describes the balance between precipitation and evaporation but does not account for *infiltration*. In addition, the dataset from Li et al. (2022) does not provide evaporation maps and we used, in this instance, precipitation maps directly as a proxy for net water flux. This partly explains why the mean precipitations in the dataset from Li et al. (2022) largely exceed those from Valdes et al. (2021).

References

Li, X. *et al.* A high-resolution climate simulation dataset for the past 540 million years. *Sci Data* **9**, 371 (2022).

Salles T. *et al.* Hundred million years of landscape dynamics from catchment to global scale. *Science*, **379**, 918–923 (2023).

Valdes, P. *et al.* Deep ocean temperatures through time. *Clim. Past.* **17**, 1483–1506 (2021).

4. And what about seasonality in precipitation – due to erosion being a non-linear function of precipitation, I would expect the seasonal precipitation to be required, in particular in monsoon regions. Again, these variables are available from the climate model.

We are aware that this information is available for the paleo-climate simulations from Valdes et al. (2021).

A challenge in modelling the climate effect on landscape is determining the importance of average or seasonal precipitation (Perron, 2017). Specially its role on soil saturation and transport. Landscape-scale models that

incorporate both background saturation and infiltration driven by rainfall time series are applied over the timescales of individual storms (Iverson, 2000) but upscaling to longer time intervals relevant to landscape evolution remains mostly limited to catchment scale (Bellugi et al. 2015).

Applying such method is above the resolution of our approach. In this study, our mesh resolution is about 5 km and the implicit time scale used to solve Eq. 1 is set to 1000 years. We have updated the Methods section to highlight this limitation in the text (line 166-169 p. 4).

While we do not account for seasonal precipitation, the approach used in the landscape evolution model is based on the stream power law which has been extensively used to predict the rate of river induced erosion over geological time (Whipple & Tucker, 1999; Tucker & Whipple, 2002; Royden & Perron, 2013; Adams et al, 2020; Salles et al., 2023). In addition, while *climate* collectively encompasses many variables (precipitation, temperature, storminess), we only consider mean annual precipitation because 1. it can be constrained from palaeoclimate records and available from paleo-simulations and 2. it influences rates of both long-term weathering (Chadwick et al., 2003) and fluvial incision (Ferrier et al., 2013).

References

- Adams, B. A. et al.** Climate controls on erosion in tectonically active landscapes. *Sci. Adv.*, **6**, eaaz3166 (2020).
- Chadwick, O. A. et al.** The impact of climate on the biogeochemical functioning of volcanic soils. *Chem. Geol.* **202**, 195–223 (2003)
- Ferrier, K. L. et al.** Climatic control of bedrock river incision. *Nature* **496**, 206–209 (2013)
- Iverson RM.** Landslide triggering by rain infiltration. *Water Resour. Res.* **36**: 1897–910 (2000).
- Bellugi D. et al.** Predicting shallow landslide size and location across a natural landscape: application of a spectral clustering search algorithm. *J. Geophys. Res. Earth Surf.* **120**: 2552–85 (2015).
- Perron, J. T.** Climate and the Pace of Erosional Landscape Evolution. *Ann. Rev. of Earth and Plan. Sci.* **45**, 561-591 (2017).
- Royden, L. & Perron, J. T.** Solutions of the stream power equation and application to the evolution of river longitudinal profiles. *J. Geophys. Res. Earth Surf.* **118**, 497–518 (2013).
- Salles T. et al.** Hundred million years of landscape dynamics from catchment to global scale. *Science*, **379**, 918–923 (2023).
- Tucker, G. E. & Whipple, K. X.** Topographic outcomes predicted by stream erosion models: Sensitivity analysis and intermodel comparison, *J. Geophys. Res.*, **107** (B9), 2179 (2002).
- Whipple, K. X. & Tucker, G. E.** Dynamics of the stream-power river incision model: Implications for height limits of mountain ranges, landscape response timescales, and research needs. *J. Geophys. Res. Solid Earth* **104**, 17661–17674 (1999).

5. Finally, it is stated that (methods, page 1, line 17) “In our formulation, the weathering impact of precipitation...”. However, weathering is also highly dependent on local temperature, and yet local temperature is not used as input to the erosion/weathering model, as far as I can tell (and yet again is available for these climate model simulations).

This is correct, we do not use the temperature in our model even though high temperatures and increase in precipitation facilitate weathering of rocks (Eppes et al., 2020).

As explained in our formulation, the weathering impact of precipitation and its role on river incision enhancement is incorporated by scaling the erodibility coefficient with local mean net annual precipitation rate. Several studies have shown that the temperature effects on weathering is of second order compared to precipitation (for example Murphy et al. (2016) for the Kohala Island).

In addition, mechanical weathering induced by expansion and contraction due to temperature variations is highly dependent on rock lithological composition (Vanwalleghem et al., 2013). In this study, we do not explicitly account for different lithologies and assume a uniform erodibility across all continents.

Integration of specific constrains from lithological information is definitely possible on a technical (algorithmic) sense. For example, we could define a method based on categorisation of lithological classes as proposed in Moosdorf (2018). However, this approach will require global paleo-lithological surficial cover that are difficult to obtain when looking into deep geological times.

We discuss the point raised by the reviewer about the impact of temperature on weathering in the Method section (line 166-169 p. 4).

References

Eppes, M. C. et al. Warmer, wetter climates accelerate mechanical weathering in field data, independent of stress-loading, *Geophysical Research Letters*, **47**, 2020GL089062, (2020).

Moosdorf, N. et al. A global erodibility index to represent sediment production potential of different rock types. *Appl. Geogr.*, **101**, 36–44 (2018).

Murphy, B. et al. Chemical weathering as a mechanism for the climatic control of bedrock river incision. *Nature* **532**, 223–227 (2016).

Salles T. et al. Hundred million years of landscape dynamics from catchment to global scale. *Science*, **379**, 918–923 (2023).

Vanwallegem, T. et al. A quantitative model for integrating landscape evolution and soil formation, *J. Geophys. Res. Earth Surf.*, **118**, 331–347 (2013).

Specific comments

1. Page 1, lines 23-37: The possible impact of sampling bias on the apparent “monotonic increase of diversity over time in the terrestrial realm” should be acknowledged and discussed in the introduction. Especially for the more recent Phanerozoic, e.g., how confident are we that e.g., the Last Glacial Maximum was more diverse than the mid Cretaceous?

We agree with the reviewer, our predicted evolution of sedimentary flux could illustrate fossil preservation bias and we have updated the text accordingly discussing this potential sampling bias in the relevant section, equally following reviewer #1 suggestions (lines 148-153 in p. 4).

2. Page 1, line 39: “The changing paleo-geography in turns modulates the atmosphere”. I do not know what “modulates the atmosphere” means. This is very unclear. A standard scientific term should be used. In addition, specific examples would help, e.g., topographic highs generating Rossby waves which influence large-scale atmospheric circulation and therefore heat and moisture transport and ultimately temperature and precipitation patterns. In addition, just as important as the atmosphere is the ocean. Paleogeographic changes affect ocean gateways and basins and therefore ocean circulation and heat transport, which ultimately can affect continental climate. There are large bodies of seminal, plus more recent, literature on this that should be cited.

We agree with the reviewer and have updated the text accordingly (lines 40 p. 1). However, we emphasize that while surely interesting, this is mostly relevant to the climate simulations, that are not the core of the current study. Describing the physics of atmospheric and oceanic circulation better belongs to the original articles, where the sets of climatic reconstructions were presented.

3. Page 3. Some checking is done of the modern results of the model compared with modern observations and of the self-consistency of the modelling framework (e.g., p3, starting line 31), and of the paleo fluxes (p3, lines 11-22), but the modelled paleo precipitations, which are of crucial importance here, are not directly validated. Again, there are several papers that have evaluated the precipitation in this climate model (and others) for various past time periods. However, the paper needs to acknowledge that modelled precipitation is highly uncertain. Model predictions vary hugely just for 100 years in the future (see e.g., IPCC AR6), let alone for 500 million years ago in the past, so this should be acknowledged and discussed.

This is a very similar comment to the major comment (2), and we believe we have now properly acknowledged and discussed these uncertainties in the previous responses and in the revised manuscript.

4. Page 4, lines 5-12: Correlation is found between sediment flux and marine organisms. There is a plausible and previously suggested mechanism for this, but still, correlation does not imply causation. This becomes even more apparent for the terrestrial, where a correlation is found between available sediment and

continental biodiversity, but this seems rather tenuous; there are many other aspects that could be playing an important role, and just because variable A and variable B increase through the Phanerozoic doesn't mean that A causes B. This should be acknowledged in the text. Even better, this is where an actual biodiversity model (e.g., that of Erin Saupe) could be used to support these ideas.

We agree with the reviewer's point. We indeed find a good correlation between model-predicted net sediment flux to the ocean and observed marine biodiversity, and between model-predicted continental sediment area and observed continental biodiversity. Based on this finding, we hypothesise that evolution of continental physiography and sediment flux might be playing an important role in the diversification of the biosphere along with other mechanisms likely critical such as biotic processes.

Running biodiversity models (from E. Saupe or others such as gen3sis see Hagen et al., 2019, 2022) based on the predicted physiography reconstructions is beyond the scope of this work but we agree that it would be a great avenue for future studies. Indeed, our group is precisely working on running such simulations, which should make the core of a PhD thesis).

References

Hagen, O. et al. Mountain building, climate cooling and the richness of cold-adapted plants in the northern hemisphere. *Journal of Biogeography* **46**, (2019).

Hagen, O. Coupling eco-evolutionary mechanisms with deep-time environmental dynamics to understand biodiversity patterns. *Ecography* **e06132**, (2022).

5. Figure 3,4: What causes this variability in modelled sediment flux and sediment area? Is it primarily coming from the time-varying modelled precipitation, or from the time-varying tectonic change, or both? Sensitivity studies in which various aspects are held constant in time would enable this question to be answered.

One of the main advantages of our approach is in its ability to estimate, for the first time, the compounding effects of these different forcings. The sediment flux and area variations are induced by 3 main factors: the imposed paleo-precipitation, the history of tectonic plates, and the imposed paleo-geography conditions (Extended Data Fig. 3).

Except for the precipitation forcing, the respective roles of each of this forcing would be difficult to quantify individually at global scale. As mentioned before, we are now evaluating the model sensitivity to initial conditions using another set of paleo-precipitation reconstruction and we present in the response to the Method comments a comparison between two paleo-geographies for the Aptian period (Fig. R3).

6. P5, line 13: "increasing physiographic variety". How robust is this, given the huge uncertainties in the reconstructed paleo geographies, especially prior to the Cretaceous where the paleo geographies are even more uncertain. What about the fact that biological evolution and biodiversity themselves cause changes in erosion rates, driven by factors such as the development of root systems, or even the evolution and spread of earthworms. This paper views the causal process very much one directional, whereas in reality the biotic and sedimentological factors are tightly coupled.

Like our responses to some of the previous comments, there are indeed many uncertainties related to the paleo-geographies themselves but also to the climates, plate reconstructions, and limitations of the landscape evolution model that we use. In Salles et al. (2023), we show how our approach could be used to quantify some of these uncertainties.

We agree with the reviewer, the biotic and sedimentological factors are indeed tightly coupled (Gibling & Davies, 2012). As explained in previous responses, we have revised the text and acknowledge some of the limitations and uncertainties in the revised Methods section of the manuscript (p. 4 to 6).

References

Gibling, M. & Davies, N. Paleozoic landscapes shaped by plant evolution. *Nat. Geosci.* **5**, 99–105 (2012).

Salles T. et al. Hundred million years of landscape dynamics from catchment to global scale. *Science*, **379**, 918–923 (2023).

7. P6, line 26: “climate control on hydrological and CO2 conditions” – this is a very odd thing to say here, because the precipitation changes (which themselves are driven by CO2 changes) are determining the predicted sediment and nutrient fluxes.

Based on reviewer’s comment, we have modified this line in the conclusion.

Methods comments:

1. Global physiography model. What spatial (and temporal) resolution is the goSPL model run at? I assume the same resolution as the paleogeographic / paleoclimatic input data, but this is not made explicit. Actually, I see that this is mentioned in the next section, but it should be mentioned here first.

Following reviewer’s comment, we are now also mentioning the spatial and temporal resolution of our simulation in the first part of this section.

2. Paleo-elevation and paleo-precipitation forcing. For the paleogeography inputs, I agree with the authors that the paleogeographies are uncertain – this is particularly true in the earlier Phanerozoic. It is mentioned that other paleogeographies could be used. Here, I would like to see some quantification of the uncertainty that this would bring to the results (e.g., by using an alternative reconstruction for one or two of the timeslices (e.g., Straume et al, 2020)).

As pointed out by the reviewer, there is no shortage of paleo-geography reconstructions that could potentially be used with our approach (Ziegler et al., 1985; Markwick & Scotese, 2004; Straume et al., 2020; Vérard et al., 2015; Poblete et al., 2021). One of the limitations of most of these reconstructions is that they are often restricted to specific time periods and rarely cover the entire Phanerozoic. Second, and maybe the most critical for our study, there is no other open-sourced global scale paleo-elevation reconstructions tied to a plate tectonic model and to a series of paleo-precipitation maps for the entire Phanerozoic. Third, only a handful are publicly made available as convenient numerical grids.

Figure R3. Impact of paleogeographies and paleoclimates on sediment flux dynamic for the Aptian period (~120 Ma). a. Predicted continental erosion and deposition maps computed using the paleo-elevation of Scotese & Wright (2018) and its associated paleo-precipitation from Valdes et al. (2021) in **a** and using the continental reconstruction from Michel et al. (2019) and paleo-precipitation obtained from the MITgcm and FOAM ocean–atmosphere general circulation numerical model (Pohl et al. (2019) in **b**. Estimated sediment flux delivered to the ocean (blue gradients circles) and stored on continents (orange circles) are shown as well as the percentage of endorheic basin area for each simulation.

Following on the reviewer's comment, we present in Fig. R3 a test for the Aptian period (~120 Ma) using a different set of paleogeography (Michel et al., 2019) and paleoclimate (Pohl et al., 2019) reconstructions. This second paleogeography is initially based on a coarser version of Scotese's paleo-maps combined with the paleogeographic reconstruction of Getech that were then modified to account for the closure of specific ocean gateways (Lunt et al., 2016). The climatic conditions are obtained using MITgcm and the FOAM models (Pohl et al., 2019).

The results in Fig. R3 highlight several differences at regional scale. For instance, a more humid equatorial climatic zone in the second reconstruction induces higher erosion rates on the northern and central part of Gondwana. The paleo-elevation differences also cause changes in the distribution of the major drainage systems and in the volumes of sediment transported to the oceans or stored in continental basins. This is the case in Antarctica and on the eastern part of Eurasia where we note higher erosion rates or an increase in terrestrial sediment accumulation depending on the considered paleogeography. Despite these local variations, when evaluating the global response (similar to what is done in our study), those disparities become tenuous. As an example, the percentage of endorheic basins varies from 24 to 26.5% between the two simulations and the net sediment flux to the ocean changes from 2.72 to 2.26 km³/yr (our estimates from the entire Phanerozoic range between 1 and 5 km³/yr).

It suggests that despite regional differences and if the imposed forcing conditions are not too dissimilar (both in terms of paleoclimates and paleogeographies), the reported global evolution and global trends that are presented in our study should remain relatively unchanged between reconstructions. However, we believe that while such tests are interesting and supportive of our approach, they remain beyond the scope of the current study, and we did not include them in the revised version.

References

Lunt, D. J. et al. Palaeogeographic controls on climate and proxy interpretation. *Clim. Past* **12**, 1181–1198 (2016).

Markwick, P. & Valdes, P. Palaeo-digital elevation models for use as boundary conditions in coupled ocean - Atmosphere GCM experiments: A Maastrichtian (late Cretaceous) example. *Palaeogeography, Palaeoclimatology, Palaeoecology* **213**, 37–63 (2004).

Michel, J. et al. Marine carbonate factories: a global model of carbonate platform distribution. *Int J Earth Sci* **108**, 1773–1792 (2019).

Poblete, F. et al. Towards interactive global paleogeographic maps, new reconstructions at 60, 40 and 20Ma, *Earth-Science Reviews* (2021).

Pohl A, et al. Quantifying the paleogeographical driver of Cretaceous carbonate platforms development using paleoecological niche modeling. *Palaeogeogr Palaeoclimatol Palaeoecol* **514**, 222–232 (2019).

Scotese, C. & Wright, N. Paleodigital Elevation Models (PaleoDEMS) for the Phanerozoic. **PALEOMAP Project** (2018).

Straume, E. O. et al. Global Cenozoic Paleobathymetry with a focus on the Northern Hemisphere Oceanic Gateways. *Gondwana Research* **86**, 126-143 (2020).

Vérard, C. et al. 3D palaeogeographic reconstructions of the Phanerozoic versus sea-level and Sr-ratio variations. *J. Palaeogeogr.* **4**, 64–84 (2015)

Ziegler, A. et al. Paleogeographic Interpretation: With an Example From the Mid-Cretaceous. *Annual Review of Earth and Planetary Sciences* **13**, 385–428 (1985).

3. Paleo-elevation and paleo-precipitation forcing. There are several issues here. First, there is a complete disconnect between the resolution of the input precipitation fields (3.75 degree by 2.5 degrees, i.e., about 270km at the equator) and the resolution that this is being interpolated to (5km globally!). In reality, precipitation patterns have a huge amount of spatial variability which is lost due to the low-resolution nature of the climate model. Due to the highly non-linear nature of precipitation-driven erosion and weathering, spatial downscaling of the climate model outputs should be applied (related to the high-resolution topography, and e.g., moisture transport direction). Secondly, as commented above, the modelled precipitation is highly uncertain, and sensitivity studies should be carried out to quantify the impacts of this uncertainty on the results.

This is a very similar comment to the major comment (2), and we believe we have addressed the scientific issues in the previous responses. In particular, we want to highlight that we have incorporated in the revised version a complete new set of simulations using the paleo-climate precipitation maps from Li et al. (2022) to evaluate the sensitivity of our results to climate.

Reference

Li, X. *et al.* A high-resolution climate simulation dataset for the past 540 million years. *Sci Data* **9**, 371 (2022).

Minor comments

1. Line 38: “dance of continents” is somewhat informal. There are many such instances of use of informal /vague language, e.g., “burst in precipitations”, “booming flux of sediments”.

Following reviewer’s suggestion, we have reviewed some of these instances in the revised manuscript.

2. Figure 1: Panel (a) looks pretty but misses out half the planet. For a scientific paper, this is better represented as a map.

We are providing maps in the supplementary materials (Extended Data Fig. 1) and movies highlighting changes in physiography and erosion/deposition rates. We are also making all our simulation outputs available as NetCDF files via a Hydroshare link.

3. Many terms are used to describe the driving forces of diversification, and to me the difference between them (if any) is unclear. Through the paper, these terms need to be defined and used consistently. For example, in the Abstract alone, the following are all used: “geological or climatic... changes in the physical environment”, “geodynamic and climatic forcing”, “Earth’s physiography”, “landscape dynamics”.

The reviewer does highlight our need to be more consistent with the terminology used in the manuscript and we have reviewed some of those terms throughout our revised version.

4. It would be very informative to see maps in the Supp Info of the input precipitation from the climate model (especially if this is downscaled to a higher resolution than the input climate model precipitation, see comments above).

We have produced a supplementary set of figures which provide the precipitation input from the climate model from Valdes et al. (2021) and the difference with Li et al. (2022) paleo-precipitation maps. This will address the request from the reviewer.

References

Li, X. *et al.* A high-resolution climate simulation dataset for the past 540 million years. *Sci Data* **9**, 371 (2022).

Valdes, P. *et al.* Deep ocean temperatures through time. *Clim. Past.* **17**, 1483–1506 (2021).

Reviewer Reports on the First Revision:

Referees' comments:

Referee #1 (Remarks to the Author):

Review of:

“Landscape dynamics and the Phanerozoic diversification of the biosphere”

Nature manuscript number 2023-04-06786A

Dijon, July 26, 2023

Alexandre Pohl (CNRS Researcher @ Biogéosciences, UMR 6282 CNRS / Université de Bourgogne, 6 Boulevard Gabriel, 21000 Dijon, France)

General comment:

During the first round of comments, I acknowledged the quality and novelty of the study by Salles et al. but asked for a number of revisions. The revised version of the manuscript satisfactorily addresses my previous comments. The authors placed the analysis of the Sr curve into the Methods, which I think makes the manuscript more straightforward. They expanded the explanation of the links that exist between the ocean nutrient inventory and marine biodiversity and, importantly, now discuss the limitations of the work and possible alternative explanations to the correlation between their simulated sedimentary flux and marine biodiversity (2 paragraphs on page 3/31 in the document with changes marked + new section ‘Limitations and sensitivity to forcing condition’ added in the Methods). A series of secondary additions is also helpful and makes the text clearer (see page 4/31 in same document). Overall, I think that the new version of the manuscript is much more pleasant and convincing and robust (notably benefitting from new sensitivity tests added in response to Reviewer #2). I agree with Reviewer #2 that many uncertainties exist in such Phanerozoic-scale study, but I think that the authors really did their best considering the current knowledge of climate and topography in the deep geological past. Therefore, I support the publication of this revised manuscript. Below, I provide minor comments.

Comments:

- Discussion of the GOBE. I realized that the most obvious model-data mismatch regarding marine biodiversity is probably that the model misses the GOBE. I think it would be worth stating that clearly in the text, and maybe to provide possible explanations (e.g., Trotter et al.1 suggested that global climate cooling may have been critical at that time).
- Extinctions. From the new discussion added in the MS, my understanding is that the results suggest that the lack of nutrients is more deleterious –as a preconditioning factor probably– to animal life than the excess? Indeed, drops in sedimentary flux predate some mass extinctions, while rises in sedimentary flux rarely coincide with extinctions. Is that well what the model suggests? It may be worth emphasizing that point since I think the general view in the community is that nutrient loading triggers anoxia and extinction. This idea would be relatively novel, in my opinion.
- Source of data in Extended Data Fig. 5b. The authors included a sensitivity test to the climate model + continental configuration (Extended Data Fig. 5). I think this is a very good addition but am a bit confused with the data used. The authors state that they used the continental configuration of

ref.2 and the MITgcm and FOAM output of ref.3. I don't remember uploading the model output for these studies, and think that the authors probably used our FOAM output for 120 hosted here (<https://doi.pangaea.de/10.1594/PANGAEA.904255>). If I'm right, then the authors should cite this paper instead⁴ (although I don't recognize the exact continental reconstruction). If this is the case, then the continental configuration would be the same as used in the baseline experiments⁵ and the climatic fields would have been simulated with FOAM. In any case, I encourage the authors to clarify this point to make sure the reader is able to track the work back to the original publications.

- Reference to sensitivity tests. I think that the authors did a great job with the new experiments, and that this additional work should be properly introduced as sensitivity testing in the main text (for now, it is mostly hidden in the supplement).

- High precipitation rates in the new simulations. In the Methods, lines 204–222 (merged document with no revision marks), the authors describe their new simulations and explain that they had to use precipitations rather than P minus E because of the lack of evaporation field in the new climatic dataset. The wording is misleading, the authors ending up writing “The higher precipitation values from Li et al”. I don't think that the precipitations are really higher. Please clarify the wording throughout.

- Wording. I suggest that the authors check throughout for small inconsistencies, e.g.,

- o Line 18: “sedimentary flux that provides”

- o Line 46: what does “its” refer to?

- o Line 119: I'm not sure I understand the reference to Fig. 3 here.

- o Line 234: “environment”

- o Methods, line 55: ref. 62.

- o Methods, line 193: delete “in”?

- o Methods, line 207: what does “their” refer to?

- o Methods, line 223: “two important information”. Pieces of information?

- o Methods, line 234: what do the authors mean with “the Phanerozoic Earth system evolution”?

References cited

1. Trotter, J. A., Williams, I. S., Barnes, C. R., Lécuyer, C. & Nicoll, R. S. Did cooling oceans trigger Ordovician biodiversification? Evidence from conodont thermometry. *Science* 321, 550–554 (2008).
2. Michel, J. et al. Marine carbonate factories: a global model of carbonate platform distribution. *International Journal of Earth Sciences* 108, 1773–1792 (2019).
3. Pohl, A. et al. Quantifying the paleogeographic driver of Cretaceous carbonate platform development using paleoecological niche modeling. *Palaeogeography, palaeoclimatology, palaeoecology* 514, 222–232 (2019).
4. Pohl, A. et al. Carbonate platform production during the Cretaceous. *GSA Bulletin* 132, 2606–2610 (2020).
5. Scotese, C. R. & Wright, N. PALEOMAP Paleodigital Elevation Models (PaleoDEMS) for the Phanerozoic (PALEOMAP Project, 2018). <https://www.earthbyte.org/paleodem-resource-scotese-and-wright-2018/> (2018).

Referee #2 (Remarks to the Author):

Just for clarification, when I stated that the paper was “potentially publishable”, I meant that it was potentially publishable in some journal, not specifically in Nature. In my view this paper presents an interesting hypothesis based largely on uncertain model outputs, and is better suited to e.g. Nature Geoscience/Communications or Science Advances – the paper that comes along and supports this hypothesis with robust geological and paleontological observations might be the Nature/Science paper.

Anyway, I commend the authors for addressing most of my comments, in particular the suggestion to use the Li et al model outputs to test sensitivity to the precipitation timeseries, and for exploring the Aptian timeslice in more detail with the Pohl et al (2019) precipitation. I note that another possibility would be to use the Pohl et al (2022) FOAM model outputs, which are available for the whole Phanerozoic, but I think that what is done here in the revised submission is (just about) sufficient in this regard.

However, there are several aspects where the authors have not sufficiently addressed my comments, in my view, and/or need improvement (before submission to some journal).

(A) In particular, as stated above, I appreciate the model precipitation sensitivity studies that have been carried out. However, this is currently hidden in Supp Info, and barely referred to in the main paper. The caveats and uncertainties described in detail in the Supp Info are critical and need to be summarised in the main paper. This could come after the paragraph beginning “Here we propose a novel method to...”. This uncertainty should also be summarised in the Conclusions, and as a single sentence in the Abstract, because otherwise the Abstract is misleading.

(B) There is some interesting work presented in the Response to Reviewers which would be good to see and be discussed in the Supp info, e.g. Figure R1.

(C1) There are some things that are unclear in the revised methods (and main paper). In particular, in one place in Methods the authors now state that it is P-E (i.e. runoff) that is used to drive the erosion model, but everywhere else “precipitation” is still used. “Precipitation” should be replaced throughout with either P-E or Runoff.

(C2) It is suggested that the higher values for erosion from the Li dataset are likely due to the fact that only Precip is used in that instance – this should be tested by a sensitivity study using just Precip from Valdes et al.

(D) In their response, the authors acknowledge that they are presenting a Hypothesis to be tested in future work. However, this does not come across strongly enough in the main paper.

(E) In my review I gave some specific examples in the Abstract where multiple different terms were used to represent that same thing, but this has not been changed. See my original Minor Comment #3.

(F) Inundation is mentioned in the Methods, but in equilibrium (which the climate field are), $P - E = \text{Runoff}$, and the inundation is net zero.

Minor comments

“Hadley Centre model⁶²” is wrongly formatted in the methods (presumably 62 should be superscript and a link to the reference).

“Inconsistencies and uncertainties of these reconstructions hold inconveniently propagate” does not make sense (in methods).

Pohl, Alexandre, Thomas Wong Hearing, Alain Franc, Pierre Sepulchre, Christopher R. Scotese, Dataset of Phanerozoic continental climate and Köppen–Geiger climate classes, Data in Brief, Volume 43, 2022, <https://doi.org/10.1016/j.dib.2022.108424>.

POINT-BY-POINT RESPONSES TO REVIEWERS

REVIEWER 1 (Alexandre POHL)

Minor comments

1. Discussion of the GOBE. I realized that the most obvious model-data mismatch regarding marine biodiversity is probably that the model misses the GOBE. I think it would be worth stating that clearly in the text, and maybe to provide possible explanations (e.g., Trotter et al. suggested that global climate cooling may have been critical at that time).

As mentioned by the reviewer, there is a time offset between GOBE and the predicted Paleozoic rise in sediment yield. We agree that dramatic changes in the Early Phanerozoic climate could have been primordial, and we also emphasize that our model relies on reconstructions (tectonics and climate) that are increasingly uncertain as time deepens. We made this clear accordingly *“The time lag between the Great Ordovician Biodiversification Event (GOBE) and the predicted increase in Paleozoic sediment fluxes (Fig. 3) could either be explained by uncertainties in the reconstructions of the climate and tectonics, or by the overwhelming effect of climate cooling³⁰ on nutrient supply”* (page 3 to 4).

Reference

Trotter, J. A., Williams, I. S., Barnes, C. R., Lécuyer, C. & Nicoll, R. S. Did cooling oceans trigger Ordovician biodiversification? Evidence from conodont thermometry. *Science* **321**, 550–554 (2008).

2. Extinctions. From the new discussion added in the MS, my understanding is that the results suggest that the lack of nutrients is more deleterious—as a preconditioning factor probably—to animal life than the excess? Indeed, drops in sedimentary flux predate some mass extinctions, while rises in sedimentary flux rarely coincide with extinctions. Is that well what the model suggests? It may be worth emphasizing that point since I think the general view in the community is that nutrient loading triggers anoxia and extinction. This idea would be relatively novel, in my opinion.

We agree and have simplified the discussion, to better emphasize that point, avoid confusion, and make it an even stronger argument (page 4 second paragraph).

3. Source of data in Extended Data Fig. 5b. The authors included a sensitivity test to the climate model + continental configuration (Extended Data Fig. 5). I think this is a very good addition but am a bit confused with the data used. The authors state that they used the continental configuration of ref.2 and the MITgcm and FOAM output of ref.3. I don't remember uploading the model output for these studies, and think that the authors probably used our FOAM outputs. If I'm right, then the authors should cite this paper instead⁴ (although I don't recognize the exact continental reconstruction). If this is the case, then the continental configuration would be the same as used in the baseline experiments⁵ and the climatic fields would have been simulated with FOAM. In any case, I encourage the authors to clarify this point to make sure the reader is able to track the work back to the original publications.

We thank the reviewer for his comment. We had forcing paleo-data sources from these simulations mixed up in our internal database. Hence there was indeed an error in the referencing of the proposed simulation in Extended Data Fig. 5b. The paleo-topography is from Nielsen, Laugié & Lettéron and has been ran using the IPSL-CM5A2 paleo-climate model (Sepulchre et al., 2020). We have changed the references accordingly in the Methods and Acknowledgments sections.

4. Reference to sensitivity tests. I think that the authors did a great job with the new experiments, and that this additional work should be properly introduced as sensitivity testing in the main text (for now, it is mostly hidden in the supplement).

As suggested by both our reviewers, we added summary sentences of the sensitivity tests. Following Reviewer 2, we first added a sentence in the Introduction: *“Sensitivity tests using alternative climatic and tectonic models (Methods) point at regional variations of surface processes while global temporal trends remain mostly*

insensitive.” and second in the Conclusion: *“Sensitivity tests illustrate how denudation rates scale with climate reconstructions, while endorheic sediment storage is chiefly controlled by paleo-elevation reconstructions.”*

4. High precipitation rates in the new simulations. In the Methods, lines 204–222 (merged document with no revision marks), the authors describe their new simulations and explain that they had to use precipitations rather than P minus E because of the lack of evaporation field in the new climatic dataset. The wording is misleading, the authors ending up writing “The higher precipitation values from Li et al”. I don’t think that the precipitations are really higher. Please clarify the wording throughout.

We agree and have clarified, equally satisfying the request from Reviewer 2, by referring to “runoff” instead of “precipitation”.

5. Wording. I suggest that the authors check throughout for small inconsistencies.

We modified the typos and misspellings accordingly.

REVIEWER 2

Specific comments

1. In particular, as stated above, I appreciate the model precipitation sensitivity studies that have been carried out. However, this is currently hidden in Supp Info, and barely referred to in the main paper. The caveats and uncertainties described in detail in the Supp Info are critical and need to be summarised in the main paper. This could come after the paragraph beginning “Here we propose a novel method to...”. This uncertainty should also be summarised in the Conclusions, and as a single sentence in the Abstract, because otherwise the Abstract is misleading.

As suggested by both our reviewers, we added summary sentences of the sensitivity tests. Following Reviewer 2, we first added a sentence in the Introduction: *“Sensitivity tests using alternative climatic and tectonic models (Methods) point at regional variations of surface processes while global temporal trends remain mostly insensitive.”* and second in the Conclusion: *“Sensitivity tests illustrate how denudation rates scale with climate reconstructions, while endorheic sediment storage chiefly controlled by paleo-elevation reconstructions.”*. We did not, however, alter the Abstract because of obvious space limitations, but also because we don’t see how the abstract could be misleading given that our sensitivity tests essentially indicate that our conclusions hold regardless of the sets of available reconstructions used (either climatic or tectonic), and last because it would be redundant with the Introduction that follows a few lines later (on top of the Extended Data, and Conclusions).

2. There is some interesting work presented in the Response to Reviewers which would be good to see and be discussed in the Supp info, e.g., Figure R1.

We appreciate this comment, and we added the Extended Data Fig. 8 and accordingly are now discussing the correlation between continental fragmentation and biodiversity. In fairness, we integrated this discussion in a short paragraph in the “Limitations and sensitivity [...]” subsection, indicating that statistics should be treated with care.

3. There are some things that are unclear in the revised methods (and main paper). In particular, in one place in Methods the authors now state that it is P-E (i.e., runoff) that is used to drive the erosion model, but everywhere else “precipitation” is still used. “Precipitation” should be replaced throughout with either P-E or Runoff.

We agree and have clarified, equally satisfying the request from Reviewer 1, by referring to “runoff” instead of “precipitation” throughout the entire manuscript (unless “precipitation” was specifically more appropriate). In practice, and as stated in the Methods, because the release from Li et al. does not contain evapo-transpiration time slices (as opposed to the release from Valdes et al.), we thus use the bulk precipitation as a proxy for runoff in the Landscape Model, which inflates its value.

4. It is suggested that the higher values for erosion from the Li dataset are likely due to the fact that only Precip is used in that instance – this should be tested by a sensitivity study using just Precip from Valdes et al.

We respectfully decline this suggestion for 2 reasons. First, as indicated in the description of the landscape model, the main equation Eq. 1 explicitly indicates that erosion scales with runoff (albeit not linearly, but with an exponent m set to 0.5). This outcome shall therefore not be considered as a proper result. Second, running a full simulation is computationally expensive, as mentioned in our initial letter (3.5 years CPU), and in our opinion, this would be inappropriate to perform such a test to obtain what Eq. 1 readily indicates, more parsimoniously. We nevertheless explicitly indicate, in the Methods, that this relationship is built in the model “*As erosion scales with runoff (Eq. 1), this inflated runoff directly impacts the global net sediment flux [...]*”.

5. In their response, the authors acknowledge that they are presenting a Hypothesis to be tested in future work. However, this does not come across strongly enough in the main paper.

In agreement with Reviewer 2, we now conclude the main text by a final sentence that reads: “[...] *to further test our hypothesis*”.

6. In my review I gave some specific examples in the Abstract where multiple different terms were used to represent that same thing, but this has not been changed. See my original Minor Comment #3.

We are surprised by this comment and unless we misunderstood, we have to disagree, as we believe we modified as much as possible. For the record, *Minor Comment #3* pointed on that aspect that “[...] *in the Abstract alone, the following are all used: “geological or climatic... changes in the physical environment”, “geodynamic and climatic forcing”, “Earth’s physiography”, “landscape dynamics”.*”. The only exception is that we first mention the “physical environment” and follow with “geodynamic and climatic forcings” for clarification. We believe this is a useful and necessary way of referring to concepts that the average reader of the Abstract may not be familiar with. The main text has been also modified to a large extent, and we occasionally retained the word “physiography”, when it could not be used as a synonym for “landscape”.

7. Inundation is mentioned in the Methods, but in equilibrium (which the climate field are), P=E=Runoff, and the inundation is net zero.

We surely agree and have clarified this misunderstanding. Using the word “flooding” could have been misleading. We do not refer to non-equilibrium. The flooding episodes (not “inundation”) that we refer to are periods of time during which large lakes and seas develop in South America, named the Pebas basin.

Minor/Technical comments

1. “Hadley Centre model62” is wrongly formatted in the methods (presumably 62 should be superscript and a link to the reference).

The reference and formatting have been fixed.

2. “Inconsistencies and uncertainties of these reconstructions hold inconveniently propagate” does not make sense (in methods).

We have removed this sentence in the new version.

Reviewer Reports on the Second Revision:

Referee #2 (Remarks to the Author):

In my view, the authors have responded to my comments in a sufficient manner.

Referee #3 (Remarks to the Author):

Nature Review-Salles et al. Landscape dynamics and the Phanerozoic diversification of the biosphere-September 12, 2023

I encourage Nature to publish the manuscript by Salles et al.

I am not a modeler and, as stated early on, their modeling addresses quantitatively issues which were earlier addressed qualitatively by others, including myself (see especially Martin, 1996, Palaios), who attempted to integrate the fossil record of biodiversity with biogeochemical and tectonic indices, among them the use of Sr isotopes as a general indicator of nutrient runoff to the oceans, interior drainage of Pangea, etc. The work by Salles et al., along with the modeling of Sharoni and Halevi cited therein, corroborates and validates much of the earlier qualitative work and may finally encourage paleontologists (among others) to seriously address the interactions of marine and terrestrial ecosystems through geologic time.

Salles et al. use sediment input to the oceans as a proxy for nutrient input, as with Sr isotopes. I'm OK with that but I do think that at times the authors tend to overstate the importance of sediment input to the oceans as the primary driver of marine diversification. Surely, there are some other forcings (as I mention below). And many nutrients are delivered to the ocean in dissolved form or dissolved particulate carbon, which later decays and releases nutrients. This may have been critical once angiosperms evolved (Martin and Servais, 2020).

This being said, I believe most of my issues are minor and I do not see them as significant flaws but only as issues requiring minor revision:

Lines 16-17 and later lines 157-159: Despite my support for the paper, I think this ignores fundamental issues of habitat (water mass stratification) and oxygen availability, among others, related to carrying capacity in the oceans. Also, there is the issue of upwelling of nutrients in relation to plate tectonics and the positions of the continents with respect to major wind systems, which Salles et al. emphasize in lines 38-46 (see, for example, Kroeck and Servais, 2022, in Earth Science Reviews, attached).

Lines 29-36: Are these statements necessary?

Line 57: Do you mean that regional variations of surface processes act on shorter timescales than "global" ones (like those of plate tectonics) so that the global trends appear to remain (or do remain) insensitive in the modeling? If so, the statement needs to be clarified.

Line 204: How about “Triassic to the Cretaceous” to parallel “during the Cretaceous and Cenozoic” and to parallel stratigraphic (temporal) succession?

I appreciate that English appears to be a second language for at least some of the authors, and that they have done their best to write as clearly as possible but there are a few awkward phrases or word usage that detract from the MS and require some polishing, among them:

Line 12: “misses to provide”. Do you mean “fails to Provide” or “does not provide”?

Line 51: “permits to jointly quantify”. How about “allows us” or “permits us”?

Line 223: Do you mean “feeding back on the landforms...”

Line 241: “levelling off”

243: Unsure as to the use of “refrained”? Do you mean “restrained” or “constrained”?

Methods:

Lines 465-469: Can you please elaborate on the reasoning behind the choice of these parameters?

Also, since sensitivity tests were run, is it feasible to give error bars on the relevant Figure’s curves to show the range of results? If not, then OK.

Finally, correlations: Was the possibility of serial correlation considered in calculating (Pearson’s) r ? Are the data normally distributed to allow the use of Pearson’s r ? If not, then Spearman’s might be more appropriate, although given the impressive tracking of the curves, I doubt that it would make much difference to the overall results.

Finally, I ask for what I hope is a small favor: the citation of Martin (1996, Palaios). This work has met with skepticism and even ridicule (to my face) by some workers. It is therefore gratifying to me that sophisticated modeling efforts are now corroborating what I saw in the fossil record and biogeochemical indices.

Author Rebuttals to Second Revision:

REVIEWER (Alexandre POHL)

1. I just have one comment: as per the discussion of the GOBE, The impact of global climate cooling on biodiversity would not necessarily involve nutrient levels. Please delete 'on nutrient supply'.

We deleted the end of the sentence.

REVIEWER 2

The authors have responded to my comments in a sufficient manner.

REVIEWER 3 (Ron Martin)

Main

1. Lines 16-17 and later lines 157-159: Despite my support for the paper, I think this ignores fundamental issues of habitat (water mass stratification) and oxygen availability, among others, related to carrying capacity in the oceans. Also, there is the issue of upwelling of nutrients in relation to plate tectonics and the positions of the continents with respect to major wind systems, which Salles et al. emphasize in lines 38-46 (see, for example, Kroeck and Servais, 2022, in Earth Science Reviews, attached).

We agree, several other processes also contribute to regulate the biodiversity. Indeed, equally valid processes were proposed by the initial reviewers or arose from informal discussions with our colleagues, and we accounted for some. However, as a comprehensive review does not fit the limitations inherent to the format of the journal, we suggest that appending adverbs (“extensively contingent” instead of “contingent”, in the main text; “strongly reliant” instead of “reliant” in the abstract) should leave sufficiently explicit the possibility that other processes come into play.

2. Lines 29-36: Are these statements necessary?

We believe these statements provide some context to the study presented here and might be useful for non-specialists. If there is a need to reduce the length of the text, we would be happy to reduce/remove these sentences.

3. Line 57: Do you mean that regional variations of surface processes act on shorter timescales than “global” ones (like those of plate tectonics) so that the global trends appear to remain (or do remain) insensitive in the modeling? If so, the statement needs to be clarified.

From our sensitivity analysis (Extended Data Fig. 5, 6 & 7), we find that there are some spatial differences in the magnitude of erosion rates and associated sediment flux. However, these differences do not significantly change the global trends (Extended Data Fig. 6b & c). We have clarified the text and added references to these Figures in the revised version of the manuscript.

4. Line 204: How about “Triassic to the Cretaceous” to parallel “during the Cretaceous and Cenozoic” and to parallel stratigraphic (temporal) succession?

We modified the text accordingly.

5. Line 12: “misses to provide”. Do you mean “fails to Provide” or “does not provide”?

We modified the text accordingly.

6. Line 51: “permits to jointly quantify”. How about “allows us” or “permits us”?

We modified the text accordingly.

7. Line 223: Do you mean “feeding back on the landforms...”

We modified the text accordingly.

8. Line 241: “levelling off”

We modified the text accordingly.

9. 243: Unsure as to the use of “refrained”? Do you mean “restrained” or “constrained”?

We modified the text accordingly.

Methods:

1. Lines 465-469: Can you please elaborate on the reasoning behind the choice of these parameters?

We explain the calibration process of these parameters at the end of the following section:

“The parametrisation of Eq. 1 is obtained by calibrating its variables with modern estimates of average global erosion rates¹⁸ (mean value of 63 m/Myr with a standard deviation of 15 m/Myr – Extended Data Fig. 6c) and of suspended sediment flux from the BQART model^{19,20} (corresponding to 12.8 Gt/yr).”

2. Also, since sensitivity tests were run, is it feasible to give error bars on the relevant Figure’s curves to show the range of results? If not, then OK.

The sensitivity tests were performed in two ways. First, we used modern estimates of global average denudation rates based on preserved sediments (Wilkinson & McElroy, 2007) and recent rates (Willenbring et al., 2013). The associated range is shown on the right-hand side of Extended Data Fig. 6c. For the second set of sensitivity tests, we show the differences between different set of climatic forcings (precipitation conditions from Valdes et al., 2021 and Li et al., 2022). Results from these two models over the Phanerozoic are presented in Extended Data Fig. 6b to c (ocean sediment flux, deposition flux and denudation rate respectively).

REFERENCES

Wilkinson, B. & McElroy, B. The impact of humans on continental erosion and sedimentation. *Geol. Soc. Am. Bull.* 119, 140–156 (2007).

Willenbring, J., Codilean, A. & McElroy, B. Earth is (mostly) flat: Apportionment of the flux of continental sediment over millennial time scales. *Geology* 41, 343–346, DOI: [10.1130/G33918.1](https://doi.org/10.1130/G33918.1) (2013).

Li, X. *et al.* A high-resolution climate simulation dataset for the past 540 million years. *Sci Data* 9, 371 (2022).

Valdes, P. *et al.* Deep ocean temperatures through time. *Clim. Past.* 17, 1483–1506 (2021).

3. Finally, correlations: Was the possibility of serial correlation considered in calculating (Pearson’s) r? Are the data normally distributed to allow the use of Pearson’s r? If not, then Spearman’s might be more appropriate, although given the impressive tracking of the curves, I doubt that it would make much difference to the overall results.

We tested the results from both Pearson and Spearman correlations. Here we chose to only show the results from the Pearson analysis as the difference between the two correlations are negligible (as pointed by the reviewer). For the relationship between the sediment flux to the oceans and diversity of marine animal families during the Phanerozoic we find a Pearson value of 0.876 and a Spearman of 0.853. For the correlation between continental sediment deposition and physiographic complexity, and diversity of vascular plants we found a Pearson value of 0.914 and a Spearman of 0.935.

4. I ask for what I hope is a small favor: the citation of Martin (1996, Palaios). This work has met with skepticism and even ridicule (to my face) by some workers. It is therefore gratifying to me that sophisticated modeling efforts are now corroborating what I saw in the fossil record and biogeochemical indices.

We have replaced reference 55 (Antell & Saupe, 2021) by Martin (1996).

REFERENCES

Martin R. E. Secular increase in nutrient levels through the Phanerozoic; implications for productivity, biomass, and diversity of the marine biosphere. *PALAIOS*, 11(3): 209–219. DOI: [10.2307/3515230](https://doi.org/10.2307/3515230) (1996).

Antell, G. S. & Saupe, E. E. Bottom-up controls, ecological revolutions and diversification in the oceans through time. *Current Biology* 31, R1237–R1251, DOI: [10.1016/j.cub.2021.08.069](https://doi.org/10.1016/j.cub.2021.08.069) (2021).